# Early Archaean subduction and intracrustal processes: experimental evidence from the East Pilbara Terrane, Australia

Alan R. Hastie [1] ✉, Sally Law[1,2], Lindsay A. Young[1] & Anthony I. S. Kemp [3]

The growth and recycling of continental crust has resulted in the modification of Earth's mantle, hydrosphere, atmosphere, and biosphere. Before the formation of the oldest stable continents, the Earth's surface was composed of 25-45 km thick basaltic crust. From 4.3-3.5 billion years ago (Ga), this basaltic crust began to differentiate, generating the first stable silicic continental crust. The tectonic processes responsible for the formation of 4.3-3.5 Ga continental crust remain controversial. Suggested explanations include deep subduction and/or crustal drip/delamination processes and a variety of relatively shallow intracrustal mechanisms. Here, we conduct high-pressure-temperature experiments on a 3.52 Ga early Earth basaltic source rock from the Pilbara Craton, Australia to show that magmas with early continental granitic compositions can form from contemporaneous shallow intracrustal and deep subduction-like tectonic environments. These results suggest that a primitive type of plate tectonics may have operated on the early Earth.

The oldest continental crust grew and stabilised in the Eoarchaean-Palaeoarchaean (4.0–3.5 Ga)[1] and although recent $^{142}Nd$ isotopic data suggest that the Nuvvuagittuq gneiss belt in Canada could be as old as 4.3 Ga[2], near-chondritic zircon εHf isotope values in Eoarchaean rocks and the rarity of inherited or detrital zircons with ages >3.9 Ga suggests that the volume of stable continental crust in the Hadean and early Eoarchaean was relatively small[3–5]. Plate tectonics explains continental crust formation on the present-day Earth, but the tectonic processes operating on the early Earth are poorly understood[6–10]. The controversy around early tectonic processes means that we do not know: (1) how the planet's crust began to differentiate; (2) how early surface environments were chemically modified during continental growth (e.g., magmatic volatile release into the atmosphere); and (3) how early continental growth influenced the development of life on our planet (e.g., the volcanic release of reduced gas species and subaerial weathering to liberate nutrient-sensitive elements and metals (e.g., P, Fe, Cu, Zn) for protein and enzyme synthesis). Furthermore, measurements of the Martian and Venusian surfaces show that, while relatively thick basaltic crust predominates, silicic rocks are also found[11–15]. For example, the discovery of granodiorite by the Curiosity rover in Gale Crater, Mars[13] and silicic(?) domes on Venus from Magellan orbital data[16]. Therefore, investigating how the oldest continental crust developed on Earth may be critical for understanding the formation of early silicic crust on other rocky extra-terrestrial bodies.

The earliest continental material is composed, predominantly, of compositionally heterogeneous tonalite, trondhjemite and granodiorite (TTG) rocks that are derived from the partial melting of hydrated metamorphosed basaltic (metabasic) rocks[17,18] followed by secondary magmatic differentiation[10,19]. Proposed tectonic settings on the early Earth range from relatively high-pressure plate-tectonic-like subduction and/or crustal drip/delamination (plutonic squishy lid)[20–23] processes (deep settings >1.4 GPa, >45 km)[23] to a variety of relatively low-pressure intracrustal processes such as crustal resurfacing[24]; crustal overturn[25] and sagduction[26] (shallow settings <1.4 GPa, <45 km). Thus, to investigate the broad tectonic process(es) operating on the early Earth we need to experimentally investigate the temperature, pressure and composition (P-T-X) of the putative source regions that gave rise to early TTG-like continental magmas.

[1]School of GeoSciences, University of Edinburgh, Grant Institute, King's Buildings, Edinburgh, UK. [2]School of Earth and Environment, Bute Building, University of St. Andrews, St, Andrews, UK. [3]School of Earth and Oceans, University of Western Australia, 35 Stirling Highway, Perth, Australia. ✉e-mail: ahastie@ed.ac.uk

The East Pilbara Terrane of the Pilbara Craton, Western Australia is an ideal location to source such a TTG-like metabasic source composition for experimentation because (1) it is regarded as a classic early Earth terrane and is commonly used to investigate early tectonic settings[9,25,27–30]; (2) it hosts extensive Palaeoarchaean granitic (*sensu lato*) and metabasic rock units[7,31,32]; and (3) the geology of the area has been used to argue for crust formation through both early plate tectonics[8,33] and intracrustal processes[7,25,34]. The East Pilbara Terrane is composed of ten composite 35-120 km diameter granitic-gneiss complexes, each made up of several generations of plutons, that are surrounded by greenstone rocks (Fig. 1)[9,27,33]. The older plutonic rocks belong to a TTG suite and younger intrusives are dominantly K-rich granites[9,33]. The Callina Supersuite represents 3.48-3.46 Ga TTG sections that are mostly found in the Shaw, Carlindi and Muccan complexes and in lower volumes in the Mount Edgar, Corunna Downs, Yule and Warrawagine domes[27,33] (Fig. 1). Slightly younger TTG of the 3.45-3.42 Ga Tambina Supersuite are also present and, like the Callina TTG, they are cross cut by younger granitic intrusions[33]. The East Pilbara Terrane also contains greenstones of the Pilbara Supergroup that comprise four 3.53-3.0 Ga sedimentary and volcanic-dominated groups. The oldest is the 3.53-3.43 Ga Warrawoona Group, with the ~3.52 Ga metabasaltic Coonterunah Subgroup at the base[9,27,35]. The East Pilbara Terrane TTG are thought to be derived from the partial melting of this Palaeoarchaean basaltic crust[9,36] and we test this hypothesis by undertaking partial melt experiments on a metabasic starting composition from the ~3.52 Ga basal section of the Warrawoona Group (Fig. 1).

## Results

Previous experimental studies exploring the generation of primitive continental material by Hastie et al. (2016; 2023) and Law and Hastie (2025) investigate Mesozoic and modern oceanic plateau source regions as analogues of the early Earth's mafic crustal surface. Historically, high-pressure-temperature experiments also investigate the generation of silicic plutonic and volcanic rocks (adakites) using modern to Neoarchean source regions (e.g., N- and E-MORB[37,38]) or synthetic materials[39,40]. Adam et al. (2012) performed experiments on a boninite composition from the Nuvvuagittuq complex of Quebec, Canada. In contrast, geochemical models suggest that Archean TTG-like rocks form by partial melting incompatible element-rich basaltic compositions, and the present study is a major and trace element experimental investigation of such a composition that is present in Hadean-Palaeoarchaean terranes[41,42]. Our high-pressure-temperature experiments use a natural Palaeoarchaean (~3.52 Ga) metabasalt representative sample from the Coonterunah Subgroup that is inferred to be a potential crustal source for East Pilbara Terrane TTG[7,31] and it is geochemically different from previous starting compositions (see Table S1 and Fig. S1, Supplementary Information). The sample is 179789 from Smithies et al.[31] that is a 'type example' of the Coonterunah Subgroup, has been used in geochemical and thermodynamic models[7] and has also been analysed for full major and trace elements and Nd-Hf radiogenic isotopes[5]. The sample is relatively unaltered for a Palaeoarchaean metabasalt with $K_2O$ and $Na_2O$ contents that are representative of the Coonterunah volcanic suite, relict basaltic texture and a loss of ignition of 2.2 wt.%[7]. The Coonterunah metabasalts commonly have large negative Nb-Ta anomalies[7], but some, like 179789, do not. However, sample 179789 has a high $TiO_2$ content (~2 wt.%), which makes it ideal for investigating the potential growth of residual rutile that may be responsible for the formation of negative Nb-Ta anomalies in any Coonterunah-derived TTG magmas. Hawkesworth and Kemp[29] demonstrate that Ce/Y and La/Yb ratios can be used to estimate crustal depths and they suggest that the East Pilbara Terrane originally had a crustal thickness of ≤30 km. In addition, the deepest crustal levels estimated for basaltic crust in the Palaeoarchaean are 40-45 km (up to 1.4 GPa)[43,44]. As such, we test an intracrustal origin for the East Pilbara Terrane TTG at a maximum pressure of 1.4 GPa (~45 km) and then further experiments at 1.6-2.2 GPa (~55-75 km) to simulate deeper (i.e., subduction/dripduction/delamination) environments. Hawkesworth and Kemp[29] use $Fe^{3+}/\Sigma Fe$ values from Archaean Pilbara volcanic rocks to suggest water contents of 1.9 wt.% in the source and the oxygen fugacity of modern to Hadean mantle and crust is considered to range from the quartz-fayalite-magnetite (QFM) to the nickel-nickel oxide (NNO) buffer[45–47]. Therefore, prior to experimentation, our starting Coonterunah metabasalt composition was equilibrated in a NNO environment in an atmosphere-controlled furnace at 1300 °C. On removal from the furnace, the glass was ground to a fine powder and ~2.0 wt.% water was added.

The experiments generate partial melt in equilibrium with residues of clinopyroxene (cpx), orthopyroxene (opx), plagioclase and titanomagnetite apart from experiments ≥2.0 GPa that have garnet, cpx, titanomagnetite and rutile-bearing residues (Fig. 2a). Amphibole is stable at temperatures of ~950-1020 °C. Garnet is stabilised ≥1.4 GPa, but at 1.4 GPa requires temperatures ≥1015 °C. The stability of garnet at 1.4 GPa and at high temperatures is confirmed in a reversal experiment. Garnet is ubiquitous at 1.8 GPa, so the reversal experiment EPTgw12 (Source Data, Supplementary Information) involved running starting composition 179789 at 1.8 GPa, 990 °C with 2.1 wt.% water for 24 hours. After 24 h the EPTgw12 experiment was quenched, the capsule recovered, and the same capsule was run again for 24 h at 1.4 GPa and 1000 °C. After quenching the second-stage of the experiment, scanning electron microscope images clearly show that the large garnets that grew at 1.8 GPa are still present at 1.4 GPa, providing evidence of garnet stability at 1.4 GPa and at high temperatures. In contrast, garnet did not grow at 1.2 GPa and 1020 °C (experimental run EPTgw20, see Source Data). Rutile was stabilised as a sub-micron-size accessory phase from 2.0-2.2 GPa (Fig. 2a). The rutile crystals are commonly too small to analyse, but a few are relatively thick (Fig. 2a) and can be analysed cleanly to give a pure $TiO_2$ composition. Rutile is not observed below 2.0 GPa, confirming previous experimental studies that demonstrate rutile is solely a high-pressure phase[48–50] and should not be predicted as an accessory phase in lower pressure thermodynamic models using natural metabasic $TiO_2$ starting concentrations and early Earth-like redox states and volatile contents.

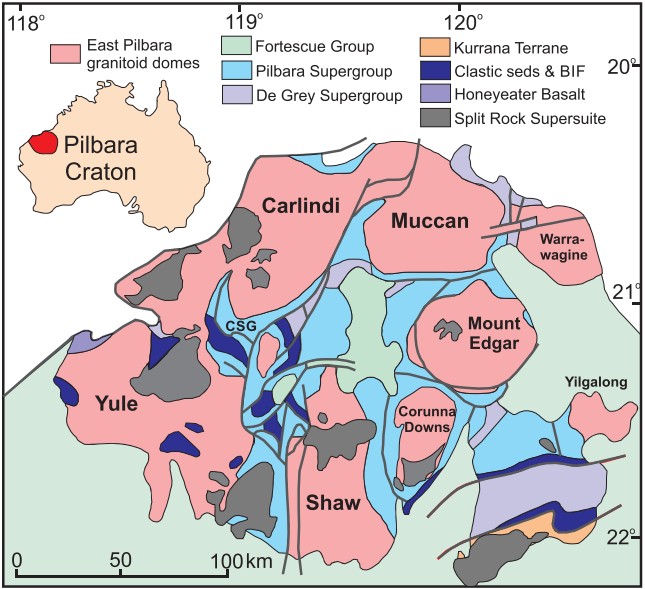

**Fig. 1 | Simple geological map of the northeast Pilbara Craton.** where major composite plutonic complexes are shown surrounded by the predominantly volcanic rocks of the Pilbara Supergroup. Location of the Coonterunah Subgroup labelled as CSG (to the south of Carlindi).

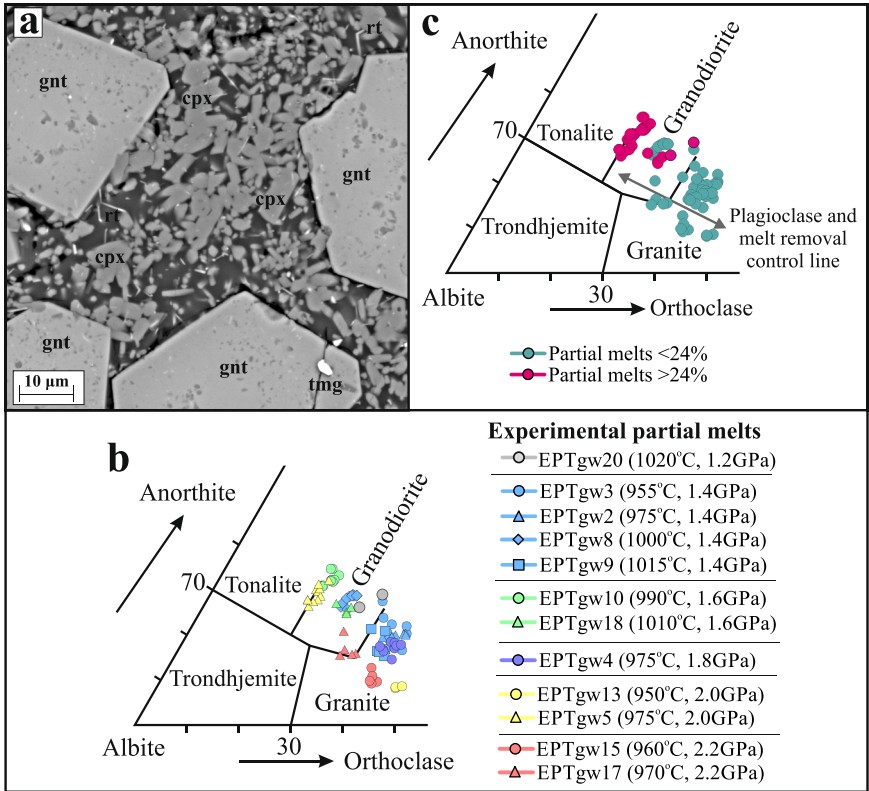

**Fig. 2 | Classification of the Coonterunah experimental partial melts.** with a representative electron-backscatter photomicrograph **a** showing residual garnet (gnt), clinopyroxene (cpx), titanmagnetite (tmg) and rutile (rt) in experiment EPTgw15 that was run at 960 °C and 2.2 GPa. **b** The analysed experimental partial melts largely plot in the granite field in the normative classification diagram of

Barker[51]. **c** Some of the experimental partial melts plot in the granodiorite and tonalite fields, but mostly those generated with degrees of partial melting above 24% (partial melts of EPTgw8 are granodiorites, but are generated with 18% fusion). Plagioclase and melt removal control line from Rollinson[10].

Partial melt compositions from our experiments are plotted on the normative An-Ab-Or ternary classification diagram[51] to show that nearly all of our experiment-derived liquids have granite and granodiorite compositions (Fig. 2b). Partial melts at 1.4 GPa have 62.7–69.8 wt.% $SiO_2$, 0.4–0.7 wt.% $TiO_2$, 14.8–18.5 wt.% $Al_2O_3$, 1.7–5.0 wt.% FeO, 0.9–2.3 wt.% MgO, 2.8–4.1 wt.% $Na_2O$ and 3.2–5.0 wt.% $K_2O$, normalised to 100% on an anhydrous basis. Deeper partial melts (≥2.0 GPa), where residual rutile is stabilised, have 65.6–71.2 wt.% $SiO_2$, 0.4–2.0 wt.% $TiO_2$ (higher due to $TiO_2$ saturation), 15.1–18.2 wt.% $Al_2O_3$, 1.7–3.4 wt.% FeO, 0.4–1.8 wt.% MgO, 3.5–4.5 wt.% $Na_2O$ and 2.2–5.5 wt.% $K_2O$. Experiments with relatively low degrees of partial melting generate granite compositions whereas higher degree partial melts have granodiorite compositions and some just plot into the tonalite field (Fig. 2c). The tonalites and granodiorites that have been formed with >24 % partial melting have 59.8–67.6 wt.% $SiO_2$, 0.4–1.1 wt.% $TiO_2$, 16.5–18.9 wt.% $Al_2O_3$, 2.8–6.2 wt.% FeO, 1.3–2.5 wt.% MgO, 2.9–4.5 wt.% $Na_2O$ and 2.2–3.4 wt.% $K_2O$. Some of the Coonterunah metabasalts have lower K contents than our starting composition[7,31] and these source rocks would potentially enable more tonalitic compositions at similar high partial melt fractions.

Trace element concentrations of the experimental partial melts are plotted on primitive mantle-normalised (pmn) multielement diagrams in Fig. 3a-b. Liquids formed at 1.4 GPa and in a lower temperature 1.6 GPa experiment (EPTgw10) have relatively flat patterns with only a slight enrichment of the incompatible elements relative to the middle and heavy rare earth elements (M-HREE) (Fig. 3a). Residual plagioclase and titanomagnetite (± amphibole) are responsible for the negative Sr and Ti anomalies, respectively. A higher-temperature run at 1.6 GPa stabilised garnet in the residue, generating liquids with low

HREE contents. All of the 1.4 and 1.6 GPa partial melts have trace element patterns unlike those for East Pilbara Terrane tonalites and trondhjemites (TT) and granites and granodiorites (GG) (Fig. 3a). Significantly, the intracrustal-like 1.4 GPa liquids do not have a substantial primitive mantle-normalised negative Nb anomaly (La/$Nb_{pmn}$ = 1.33–1.65) or low M-HREE concentrations. Importantly, partial melts formed at 1.8 and 2.0 GPa have similar trace element concentrations to the higher temperature 1.6 GPa analysis, with residual garnet generating low M-HREE compositions like those of Palaeoarchaean TTG (Fig. 3b). The lack of residual plagioclase ≥2.0 GPa prevents a pronounced negative Sr anomaly from forming. Conversely, experiments from 2.0-2.2 GPa have no residual plagioclase, but have residual rutile and garnet to generate trace element patterns more similar to East Pilbara granitic rocks, especially the beginning of the development of a strong negative Nb anomaly (2.2 GPa: La/$Nb_{pmn}$ = 1.96–2.53) (Fig. 3b).

## Discussion
Our experiments show that direct partial melting of Coonterunah metabasalt protoliths that lack a pre-existing negative Nb anomaly generates predominantly granite and granodiorite compositions, which occur in the East Pilbara Terrane plutonic centres[9]. Higher degrees of partial melting enable magmas to develop tonalite compositions, especially if the source rock has lower K, and this could explain why Pilbara tonalites and trondhjemites are volumetrically minor, considering the plutonic centres as a whole. Formation of residual garnet at ≥1.4 GPa imposes high La/Yb on our experimental partial melts (Fig. 3b), but crucially, formation of residual rutile at ≥2.0 GPa imposes a negative Nb anomaly on the experimental partial

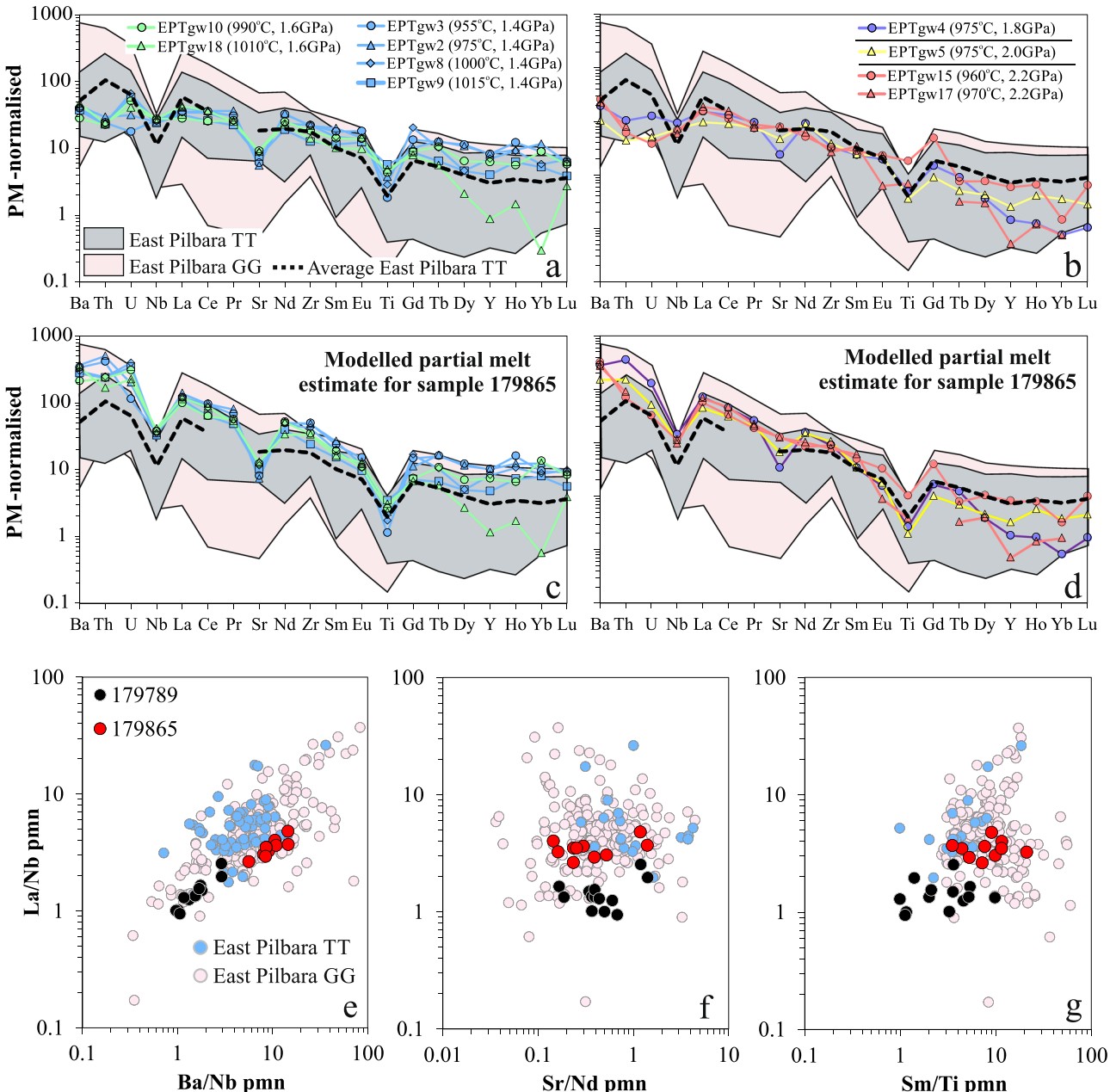

**Fig. 3 | Partial melt trace element compositions from our experiments plotted on primitive mantle multielement and ratio-ratio diagrams. a**, **b** ion microprobe analyses of our experimental partial melts from (**a**) 1.4-1.6 GPa, and (**b**) 1.8-2.2 GPa. Modelled liquid compositions using sample 179865 are shown in (**c** and **d**). Sample data from 179789 and 179865 shown in (**e**–**g**) and compared directly to East Pilbara Terrane granitic rocks are from Hawkesworth and Kemp[29]. Primitive mantle normalising values from McDonough and Sun[59].

melt, demonstrating East Pilbara TTG magmas can only be directly formed from Coonterunah source regions that lack a pre-existing Nb anomaly at pressures ≥2.0 GPa. However, some of the Coonterunah metabasalts have pre-existing negative Nb anomalies[31] and these could potentially undergo partial melting at lower intracrustal-like pressures to generate silicic magmas with negative Nb anomalies <1.4 GPa. This scenario can be modelled using our experimental results. Shaw[52] showed that the compositional mass balance relationship between a starting protolith composition and a resultant partial melt composition is described by the equation:

$$\frac{C_l}{C_0} = \frac{1}{D + F(1 - P)}$$

Where $C_l$ is the concentration of some element in the resultant partial melt, $C_O$ is the concentration of the protolith before partial melting, $F$ is the mass fraction of melt generated, $D$ is the bulk partition coefficient prior to partial melting and $P$ is the average of the partition coefficients weighted by the proportion contributed by each phase to the partial melt. We use sample 179789 as our experimental starting composition and with our experimental partial melt analyses we know the terms $C_l$ and $C_O$ in each of our experimental runs. By knowing the left-hand side of the equation, the right-hand side of the equation can be reduced to a single variable ($X$). Assuming Henry's law behaviour in the trace elements and similar modal mineral abundances in all potential Coonterunah metabasalt protoliths, we propose that $X$ is similar for all Coonterunah samples, given the specific conditions in our experimental runs. If so, $\frac{C_l}{C_0} = X$ and we can rearrange to calculate the partial

melt composition from any Coonterunah sample by $C_l = C_O X$. Coonterunah sample 179865 has a strong pre-existing negative Nb anomaly ($La/Nb_{pmn}$ = 2.44 relative to 0.89 in 179789) and if we use this sample as a source protolith, the modelled trace element partial melt compositions using our experimental conditions are shown in Fig. 3c and d. Modelled 1.4-1.6 GPa partial melts have a strong negative Nb anomaly, and the remaining trace element pattern can explain some of the Pilbara granitoids with high M-HREE contents (although, the experimental partial melt negative Sr anomalies are very pronounced). Again, the exception is the higher temperature 1.6 GPa experiment that stabilises residual garnet and is similar to Pilbara plutonic rocks with lower M-HREE concentrations (Fig. 3c). Furthermore, with high incompatible element contents, negative Nb and Ti anomalies and low M-HREE, modelled liquids from 1.8-2.2 GPa match Pilbara TT compositions very well (Fig. 3d). The similarity of the experimental and modelled Coonterunah-derived partial melts to East Pilbara Terrane plutonic rocks is clearly seen in Fig. 3e–g. $(Ba/Nb)_{pmn}$, $(Sr/Nd)_{pmn}$ and $(Sm/Ti)_{pmn}$ ratios highlight the development of negative Nb, Sr and Ti anomalies in our partial melts and it can be seen that our liquid compositions overlap with East Pilbara Terrane granitic data.

The intense controversy over the early Archaean tectonic setting of the East Pilbara Terrane focuses on the formation of the dome and basin (keel) structure (Fig. 1). The most common model suggests that dome and basin structures are attributed to vertical intracrustal mechanisms[7,34]; however, in a second model, some authors note that domes and basins can form in modern convergent margins[8,33]. Our experiments suggest that a deep tectonic environment is required, to some degree, to form the East Pilbara Terrane granitic rocks. In order to generate relatively low HREE contents and negative Nb anomalies the granitic rocks in the plutonic centres formed by either (1) direct partial melting of a subducting slab or foundering crust composed of Coonterunah-like basalts with no pre-existing negative Nb anomaly at ≥2.0 GPa to stabilise garnet and rutile in the residue and/or (2) generation of Coonterunah basaltic magmas with pre-existing negative Nb anomalies that ascend and form the Pilbara crust which subsequently re-melts in intracrustal environments ≤1.4 GPa to generate some of the Pilbara granitic compositions. Both these petrogenetic scenarios are supported by similar εNd and εHf values for Palaeoarchaean East Pilbara TTG and the Coonterunah basalts[5]. The existence of several co-existing tectonic settings is also supported in geodynamic models[21,22,53]. Higher degrees of partial melting of a basaltic protolith would then favour more tonalitic compositions.

Critically, rutile and garnet are only stabilised together at pressures ≥2.0 GPa during fusion of a hydrated Coonterunah-like metabasalt, which is deeper than the proposed East Pilbara Terrane crustal thickness of ≤30 km[29]. Consequently, only deep subduction and/or crustal drip/delamination (plutonic squishy lid)[20–23] tectonic environments can directly generate Pilbara TTG-like magmas by partial melting of a Coonterunah metabasic source region that lacks pre-existing Nb-Ta anomalies. As such, some form of subduction may have been underway on the surface of the Earth by at least the Palaeoarchaean. If Palaeoarchaean oceanic crust was subducted, new crust must also have been formed elsewhere on the planet's surface to lead to the rapid development of primitive plate tectonics. For the Pilbara Craton, primitive plate tectonics could have involved shallow underthrusting; however, the presence of Coonterunah metabasalts with pre-existing negative Nb anomalies suggests that a metasomatised mantle wedge may have existed, and that early Archaean subduction could have been relatively steep. Potential Pilbara TTG melts, derived from a slab, can easily ascend without incorporating mantle material (developing a 'mantle-signature') if (a) previous slab melts 'armour' ascending melt pathways into the lower crust to prevent or limit slab melt hybridisation and/or (b) ascending TTG magmas undergo fractional crystallisation to lower MgO, Ni and Cr contents. In addition, intracrustal processes also probably operated in the same region because low-

pressure partial melting of Coonterunah metabasic rocks with pre-existing Nb anomalies can also generate granitic rocks in the East Pilbara Terrane. Thus, we propose that both deep subduction-like and shallow intracrustal processes generated Earth's oldest surviving continental crust.

As the continents grew through both subduction zones and intra-crustal processes and stabilised throughout the early to mid-Archaean, accompanying volcanism and weathering released redox-sensitive elements into the oceans, and H, S, C, and O into the oxygen-free primitive atmosphere. These elements are critical to the development and evolution of life, which evolved under chemical conditions where redox-sensitive transition metals and volatiles became an irreplaceable component of prebiotic molecules, proteins and enzyme systems[54,55]. Thus, early Archaean subduction, intraplate partial melting and continental growth were responsible for modifying the early Earth's interior and crustal surface and are pivotal to our understanding of how the whole Earth system evolved into our modern world.

## Methods

### High pressure-temperature experiments
Experimental runs were carried out on a ½ inch (1.27 cm) end-loaded piston cylinder press at the Experimental Geoscience Laboratories, School of GeoSciences, University of Edinburgh, U.K using talc-Pyrex®-graphite assemblies (Fig. S1, Supplementary Information). Capsules were centred in assemblies using alumina spacers, within 1 mm of the thermocouple junction. Experiments were performed using the hot-piston-out technique in which runs were pressurised to 110% of the reported final pressure, heated, and pressure allowed to bleed off to the desired run pressure over 2–5 mins. Pressure was monitored and maintained during runs within 0.25 kbar of the reported pressure, and a 15% correction to the nominal pressure was applied to compensate for pressure loss due to internal friction, previously calibrated for the assembly based on the quartz-coesite transition, the kyanite-sillimanite transition, the jadeite-albite-quartz reaction, and the melting point of diopside. Temperature was measured with a Pt-Pt13%Rh thermocouple. No correction for the effects of pressure on thermocouple EMF was applied. Temperature gradients across the sample are small over the temperature range used here, in the order of 10 °C, and the temperature during runs did not deviate more than 5 °C from the reported values. Experiments were quenched by shutting off power to the heating circuits, and fell to below 300 °C after 5 seconds, and to below 100 °C after 12 seconds.

The starting material was finely-powdered glass of sample 179789 from Smithies et al.[31]. Sample powder of 179789 was produced using an agate ball mill from hand-picked rock chips rinsed in de-ionised water. The powder was glassed in an atmosphere-controlled Deltech vertical tube gas mixing furnace. A Pt crucible was pre-contaminated with 179789 material at 1300 °C in order to saturate the crucible with Fe. The initial 179789 material was removed from the crucible and -1 gram of new powder was added. This new powder was fused in the furnace at 1300 °C for 30 minutes in a stream of $CO_2$-$H_2$ gas. In line Bronkhorst Ltd mass flow controllers were used to accurately control and monitor gas flow. Deines (1970) tables were used to determine gas proportions required to anneal the sample at an oxygen fugacity close to that of the Ni-NiO (NNO) buffer, calibrated at run conditions using a zirconium dioxide oxygen sensor. The procedure aimed to limit the time available for Fe loss to the capsule and volatile element (e.g., K) loss to the furnace atmosphere. The synthesised 179789 glass was removed from the crucible and was crushed and finely powdered. Sample 179789 is a rare metabasalt and to ensure material was not potentially lost due to unforeseen circumstances, the material was glassed in three batches: EPT2019, EPT2021 and EPT2024. The glass powders were imaged on a scanning electron microscope to ensure that the material had fully glassed. Glass fragments were analysed (25-30 spot analyses) for major elements on a Cameca SX100 electron microprobe at the UoE.

Synthetic 179789 glass major element compositions are near identical to the original 179789 powder and showed no Fe, Na or K loss (Table S1, Supplementary Information). It should also be noted that the new Pilbara starting material is compositionally distinct from oceanic plateau starting materials from Hastie et al.[23,56] (1187-8 and 1187-10) that were originally analysed by Fitton and Godard[57]. EPT2019, 2021 and 2024 trace element compositions were analysed on a NERC ion microprobe at the UoE. Like the major element analyses, synthetic 179789 glass trace element concentrations are near identical to the original 179789 powder (Fig. S2, Supplementary Information).

Approximately 0.035-0.05 grams of starting material, and ~2 wt.% 18.2MΩm deionised water were loaded in welded $Au_{75}Pd_{25}$ capsules and run for 24 hours. The run has to be long enough to grow crystal phases of sufficient size for imaging and analyses; however, the longer the experimental run, the higher the chance of hydrogen escape from the capsule, which will alter the oxygen fugacity. It has long been recognised that 24 h is a good time to grow large enough phases and to avoid damaging hydrogen loss. The experiments are designed to simulate partial melting of potentially wet basaltic lid and hydrous subduction-derived material and thus require an oxygen fugacity ($f$O2) close to nickel-nickel oxide (NNO) mineralogical equilibrium. However, experiments cannot be buffered using an NNO assemblage because Ni reacts with the Pd-bearing capsule material. Nevertheless, although not explicitly buffered, the talc-Pyrex® experimental assembly used in this study imparts conditions within the sample volume close to the NNO oxygen buffer, consistent with Fe XANES measurements of silicate glass in other quenched run products (Stokes et al.[58]). Furthermore, as synthetic glass starting materials were pre-annealed at $f$O$_2$ close to the NNO buffer and only minimal changes in $f$O$_2$ are expected during our partial melting experiments. Many previous fluid-absent experiments in the literature do not approach equilibrium, but the use of our synthetic glasses enables us to do so. Equilibrium conditions in our experiments are shown by (1) all mineral phases being homogeneous and un-zoned and (2) similar phase proportions and compositions being derived from previous experiments run for 24 and 48 hours at the same pressures and temperatures (Hastie et al.[23,56]). Experimental run EPTgw20 confirms that garnet does not grow at 1.2 GPa and 1020 °C (Fig. S3, Supplementary Information).

## Major element analyses

Experimental charges were encased in epoxy resin, ground down and polished for imaging and major and trace element analysis. Samples were carbon-coated and imaged on a Zeiss Field Emission Gun Environmental SEM at the UoE. Major element and high concentration trace element analyses of run products were determined at the UoE using a five-spectrometer Cameca SX100 electron microprobe instrument with 15 kV acceleration voltage. Mineral products were measured using a fully focused beam and two conditions of 4 nA (major elements) and 100 nA (minor and trace elements). For glass analyses, a defocused beam of 2 nA and 5 μm was used for major elements to avoid Na loss and to target the relatively small melt pools, and 80 nA at 5 μm was used for minor and trace elements.

## Trace element analyses

Trace element analyses on the glasses were determined on a secondary ion mass spectrometer (SIMS) at the UoE. SIMS analysis was performed on a Cameca IMS-7f ion microprobe. Analyses were made using a 16 O$^-$ primary beam of 15 keV impact energy. A 1 nA beam was focused to a 5–8 micron spot, and 4.5 keV positive secondary ions were measured. Molecular ion overlaps were significantly reduced by the use of energy filtering (ion energies between 55 and 95 eV measured). Corrections were made for the overlap of the light rare earth elements (REE) on the heavy REE and HoO on Ta. Concentrations were determined using a combination of STHS-1, GSD-1G and SRM610 glass standards.

## Data availability

All data available in Source Data and Supplementary Information. Source data are provided with this paper.

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

## Acknowledgements

The experimental work was funded through a Leverhulme Research Grant (RPG-2019-282), NERC E4 DTP number 2894712 and the NERC standard grant VIPER (NE/X001334/1). Hugh Smithies provided rock chips of Coonterunah basalt GSWA179789.

## Author contributions

A.R.H. developed the initial idea, ran half of the experiments, imaged most of the experiments on an SEM, did half of the EPMA and SIMS analyses and wrote most of the paper. S.L. ran several experiments, imaged the runs on an SEM, analysed half the samples on the EPMA and SIMS and contributed to the text. L.A.Y. ran, imaged and analysed sample EPTgw20 to confirm a garnet-free residue. A.I.S.K. advised on, and supplied, the starting material and wrote part of the text.

## Competing interests

The authors declare no competing interests.

## Additional information

**Supplementary information** The online version contains
supplementary material available at

Alan R. Hastie.

**Peer review information** *Nature Communications* thanks Jacob Mulder,
Ali Polat, Chao Zhang, and the other, anonymous, reviewer for their
contribution to the peer review of this work. A peer review file is avail-
able.

