## [Transparent Peer Review file · Nature Communications]

Early Archaean subduction and intracrustal processes: experimental evidence from the East Pilbara Terrane, Australia

Corresponding Author: Dr Alan Hastie

Version 0:

Reviewer comments:

Reviewer #1

(Remarks to the Author)

General comments

This is a great and timely study addressing the origin of Paleoproterozoic continental crust in East Pilbara, northwestern Australia. The study addresses first order, fundamental questions in Earth science related to the origin and tectonic setting of early continental crust. The authors use major and trace element geochemical data to constrain the formation depth of granitic melts that were generated by well-constrained experiments. The authors provide a good introduction to the study topic. Objectives are clear. Arguments presented in the manuscript agree with the data reported in the manuscript and the data in the literature. The results of the study show that most of the assumptions made on the geology of East Pilbara by many previous studies are not supported by robust experimental data. So, these assumptions need to be revised. The topic of the manuscript is of great interest to the readers of Nature Communications. I strongly recommend the publication of the study as it is. It will be a well-cited paper. After the publication of this paper, geologists will look at the origin of Archean continental crust in East Pilbara from a totally new perspective.

The East Pilbara Archean terrain represents one of the key regions that is used to test Archean tectonic models. Both plate tectonic and non-plate tectonic models are proposed to explain the rock record in East Pilbara. Because of its dome and basin structure, the East Pilbara has been at the center of geological debate on the origin of Archean continental crust for many decades.

The authors conducted high pressure-temperature experiments on 3.52 Ga old mafic rock samples from the Coonterunah Subgroup to test the models proposed for the origin of the granitic rocks in the East Pilbara terrain. Experiments are described in detail. The experiments were conducted under various pressure (1.4 to 2.2 GPa) and temperature (955-1015 °C) conditions to simulate the depths of melt generation. Experiments at high pressure conditions produced melts whose major and trace element compositions match those of the granodiorites and tonalites in East Pilbara.

Specific comments

Line 1: I recommend replacement of “early Archaean” with “Paleoproterozoic”. This is up to the authors.

Lines 35-37: I am not aware of any field data providing evidence for “drips/delamination”.

Line 44: Add “crust” after “continental”.

References need to be edited.

Reviewer #2

(Remarks to the Author)

[Editorial Note: Please also see attachments at end of file]

This study by Hastie et al. conducts high-pressure-temperature petrological experiments on a well-known lithology within the Archean igneous geochemistry community in an attempt to understand the tectonic processes giving rise to Earth's first continents. While the dataset could be used to generate partition coefficients and inform future phase-equilibrium modelling, I cannot recommend it for publication in Nature Communications; the manuscript does not meet the journal's threshold for conceptual advance and breadth of interest. The novelty is narrow because it substantially overlaps with Hastie et al. (2023), Law et al. (2024) and Law and Hastie (2025), without a clear additional and substantial advance. Several core inferences rest on evidence that is currently insufficient or internally inconsistent, and the broader geodynamic claims extend beyond what the new data can support. Therefore, this contribution appears incremental for this venue and would be better suited to a specialist journal after substantial further development.

Major comments:

The cornerstone of the ≥ 2.0 GPa requirement is the appearance of rutile. In the draft, rutile is "sub-micron," identified via acicular habit and a TiO_2 spike on EDS, and explicitly stated to be too small to analyze. No Raman, micro-XRD, or TEM confirmation is provided, and no modal estimates are reported. Given that the Nb argument hinges on rutile, this evidence is not adequate. The authors state in lines 124-126 the presence of titanomagnetite in the residues of all experiments; thus, ilmenite and titanomagnetite exsolution cannot be excluded, given the inadequate evidence. I request that the authors provide definitive phase identification, otherwise, the pressure threshold for Nb depletion is not demonstrated.

The claim that garnet is unstable at 1.2 GPa and 1015 °C is unsupported; no 1.2 GPa experiment is reported in the SI. Perhaps the 1.8 GPa experiment should be re-run at 1.2 GPa, as it was at 1.4 GPa? Given that the 24-hour duration of the runs seems a little short, this might help clarify this issue. This is particularly notable because the phase equilibrium models of Johnson et al. (2017) predict garnet being stable down to a pressure of 0.8 GPa for the same general lithology (CF2, albeit an average of more samples).

I find the choice of samples lacking negative Nb-Ta anomalies difficult to reconcile if the authors' goal was to perform experiments on a lithology representative of the broader CF2 suite. The manuscript acknowledges compositional diversity within the Coonterunah metabasalts, including pre-existing Nb troughs. As demonstrated in Johnson et al. (2017), average Coucal Basalt has a negative Nb-Ta anomaly, so this calls into question whether the experiments are actually applicable to the stratigraphic unit. The Nb-Ta trough already existing in the average composition of the unit would suggest that, as a whole, rutile isn't necessary to generate a negative Nb anomaly. Without at least a second protolith run, generalization remains speculative.

The reduction of the Shaw (1970) relationship to $(C_F = C_0^X)$ by folding F, P and D into a single constant X requires similar modal mineralogy and partitioning across different Coonterunah protoliths. That assumption is strong, and it is not validated. The modelling is then used to argue that a Nb-depleted source can reproduce a Nb trough at 1.4–1.6 GPa, which undercuts the headline claim that TTG-like Nb requires ≥ 2.0 GPa. I suggest the authors either experimentally test the model with at least one run on the Nb-depleted sample (e.g., 179865) at 1.4–1.6 GPa, or significantly soften the inference. As written, the logic is circular.

The authors argue that the granitic and granodioritic compositions of the experimentally-derived partial melts can be reconciled with natural observations because Paleoproterozoic tonalites and trondhjemites in the Pilbara "represent only a very small volume of the granitic complexes, whereas granites and granodiorites are the dominant silicic plutonic rock-type." This appears to be a misunderstanding. The granitic complexes of the EPT contain multiple supersuites, and the later ones, particularly from Emu Pool onwards, contain increasingly more granites and granodiorites. The granites in the younger Paleoproterozoic supersuites have been demonstrated to be the result of partial derivation from re-melting of older TTG supersuites (e.g., Gardiner et al., 2017; Champion and Smithies, 2019; Hickman, 2021). The same goes for the granites in the Mesoproterozoic supersuites of the EPT. On the other hand, the Callina and Tambina supersuites, which are subordinate in area to the younger supersuites, are only 31% and 7% comprised of granite and granodiorite, respectively (using the compilation of Vandenburg et al., 2023, which incorporates all currently available analyses from the Pilbara). On the other hand, tonalites and trondhjemites comprise 30 and 31% of the Callina and Tambina supersuite samples (see attached Figure 1). While there might be some sampling bias here, owing to the sparsity of outcrop and the inability to sample significant portions of some of the domes due to heritage sites, the areal extents of granitoid lithologies in the two supersuites, as mapped by the Geological Survey of Western Australia, are consistent with granites being highly subordinate to tonalites and granodiorites (see attached Figure 2). While there are some examples of granites in these supersuites (e.g., the Homeward Bound Granite in the Callina Supersuite), these low-degree melts are relatively rare.

The authors imply that a wet environment would be best explained in a convergent margin. However, all that can really be shown here is that the fluid-present or fluxed melting of metabasalts indeed likely contributes to their production (e.g., Hernández-Urbe, 2024; Pourteau et al., 2020), but subduction is not a prerequisite for this style of intracrustal melting in the Archean (Hartnady et al., 2022), or in the Phanerozoic, for that matter (e.g., Weinberg and Hasalová, 2015). Numerous recent works have demonstrated that the dehydration of mafic-ultramafic lithologies in the crust can release sufficient water to produce TTGs (e.g., Hartnady et al., 2022; Hernández-Urbe, 2024; Pourteau et al., 2020; Tamblyn et al., 2023). Studies that infer a subduction-related petrogenetic origin for TTGs based on evidence for fluid-fluxed melting (e.g., Ge et al., 2023) only demonstrate that this melting process occurred within the crust. I'm not saying that TTGs can't be produced in subduction zones, more that they aren't diagnostic of the presence or absence of them.

The crustal thickness estimates that the Authors invoke as evidence for subduction-like processes are based on La/Yb, which is only one geochemical parameter. How would this compare to another parameter, such as Eu/Eu* in zircon, or even better, a method that integrates multiple elemental and isotopic proxies (e.g., Luffi and Ducea, 2022)?

TTGs are, by definition, melts of pre-existing basaltic crust that have not interacted with mantle peridotite (e.g., Smithies et al., 2000; Moyen, 2020). Tonalites and trondhjemites formed directly by subduction-related processes would have to be derived from the slab, inevitably reacting with mantle wedge peridotite during ascent, thereby becoming high-Si adakites (and producing reciprocal residues that subsequently melt to produce sanukitoids) before they can be emplaced in the crust (Martin et al., 2009; Rapp et al., 2010; Smit et al., 2024; Smithies, 2000). I think the authors need to provide a mechanism with which TTGs generated from the slab would be able to ascend to the crust without contamination from the mantle if they want to pursue that interpretation. The authors do not engage with recent work illustrating that many processes obfuscate the traditional interpretation/division of TTGs into LP-HP-MP categories, such as interstitial melt loss (Laurent et al., 2020), MgO content of the starting protolith (Johnson et al., 2017), mineral segregation (Kendrick and Yakymchuk, 2020; Kendrick et al., 2022), degree of melting and fractional crystallization (Smit et al., 2024), and melt hybridization (Hernández-Montenegro et al., 2021).

The planetary angle and broad geodynamic statements are not anchored by new data in this study and are read as overreach. Plate tectonics is a very specific regime (see e.g., Cawood et al., 2022; Nebel et al., 2024). Neither drips/delamination nor subduction-like processes imply a global network of discrete plates separated by convergent, divergent, and transform boundaries with movements relative to one another governed by slab pull and minor amounts of ridge push. Such features can also occur in other types of non-plate tectonic regimes, such as sluggish lid (Lenardic, 2018), squishy lid (Lourenço et al., 2020) or lid-and-plate regime (Capitanio et al., 2019). I suggest keeping the focus on what the experiments demonstrate and moving speculative implications to a restrained paragraph.

The authors state on line 120 the equilibration of the compositions at a NNO environment, but in the methods state "However, experiments cannot be buffered using an NNO assemblage". This seems misleading to me and should be clarified in the main text, as non-specialists might think the experiments are being run at NNO.

Line-by-line suggestions:

Line 66: unlike all the other modes listed here, sagduction is entirely internal.

Line 72: I think a more detailed map figure might be warranted here; in its current state, it doesn't provide much context other than the location of the CF2 site.

Line 79: Following up on my comment on line 66, I'd argue that a study like Vandenburg et al. (2023) deals more with the tectonic evolution of the craton than Francois et al. (2014) and would be a more appropriate reference here.

Lines 89-90: Almost all the supersuites in the EPT are intruded by the Split Rock Supersuite, so this information isn't germane here.

Line 93: Smithies et al. (2005- Whundo) isn't really an appropriate reference here. Perhaps you're thinking of "It started with a plume..." by Smithies et al. (2005)?

Line 94: Not entirely; see Hickman (2023) for a recent overview of evidence for Eoarchean crust.

Lines 101-103: I don't see how alteration would be an issue; if anything, studies such as André et al. (2019, 2022) and Murphy et al. (2024) suggest that alteration is an important aspect of TTG sources.

Line 112: Chowdhury et al. (2021) estimated a crustal thickness of ~50 km at 3.35 Ga for the Singhbhum Craton. See also Figure 6 in Chowdhury et al. (2025).

Fig. 3: Specify the MORB normalization dataset; it is ambiguous as to which study the normalizing factors are from. Likewise, N-MORB is a surprising choice of normalizing factor for Archean rocks, as N-MORB did not exist during the Archean (Barnes et al., 2021). I think normalizing to primitive mantle would be a better choice.

Fig. 3: The shaded areas denoted "East Pilbara TT" and "East Pilbara GG"; are they the full range of trace element data for compiled granitoids? They're so broad that they are not discriminating and don't really impart much meaningful information. Perhaps a better idea would be to plot the mean, median, or geometric means bounded by something like the 25th and 75th percentile values?

Fig. 3: Trace element ratio-wise, melts from 179989 don't seem to be very representative of Pilbara granitoids.

Lines 189-190: The authors state "All of the 1.4 and 1.6 GPa partial melts have trace element patterns unlike those for East Pilbara tonalites and trondhjemites (TT) and East Pilbara granites and granodiorites (GG) (Fig. 3a)." Yet, the way that the figure is presented currently suggests that the partial melts actually do have trace element patterns similar to EPT granitoids.

Line 203: I believe this is a misinterpretation once again of Hickman (2004), where the line about granitoids being "dominantly monzogranite and granodiorite" is in reference to the granitoids of the Pilbara Craton as a whole, rather than the

EPT, let alone the Callina and Tambina Supersuites. As I have discussed above, the authors' assertion that granites and granodiorites are "volumetrically dominant in the East Pilbara plutonic centres" is not germane to the central comparison, given that this volumetric dominance is the consequence of later magmatism incorporating an increasing amount of reworked components, rather than due to conditions during the generation of TTGs of the Callina and Tambina Supersuites.

Lines 256-258: The majority of the evidence does not support this inference (Hickman et al., 2021, 2023), at least not until ca. 3.25 Ga, ca 200 Myr after emplacement of the Callina and Tambina Supersuites.

Lines 273-276: One wouldn't expect the eclogitic residue to be observed in outcrop. The TTGs were emplaced into the middle crust (e.g., Wiemer et al., 2018; Champion and Smithies, 2019), and there are no sections of lower crust exposed in the Pilbara. Given the extreme density contrast between purported eclogites and felsic melts, I don't see how a scenario where TTGs carried eclogite fragments during migration from the lower to the middle crust would be physically possible (i.e., Stokes' Law). I also cannot recall, to the best of my knowledge, an example of such a phenomenon occurring in Phanerozoic/modern localities associated with crustal delamination.

References:

- André L., Abraham K., Hofmann A., Monin L., Kleinhanns I. C. and Foley S. (2019) Early continental crust generated by reworking of basalts variably silicified by seawater. *Nat. Geosci.* 12, 769–773.
- André L., Monin L. and Hofmann A. (2022) The origin of early continental crust: New clues from coupling Ge/Si ratios with silicon isotopes. *Earth Planet. Sci. Lett.* 582, 117415.
- Barnes S. J., Williams M., Smithies R. H., Hanski E. and Lowrey J. R. (2021) Trace Element Contents of Mantle-Derived Magmas Through Time. *J. Petrol.* 62.
- Capitanio, F.A., Nebel, O., Cawood, P.A., Weinberg, R.F., Cloc, F., 2019. Lithosphere differentiation in the early Earth controls Archean tectonics. *Earth Planet. Sci. Lett.* 525, 115755.
- Cawood P. A., Chowdhury P., Mulder J. A., Hawkesworth C. J., Capitanio F. A., Gunawardana P. M. and Nebel O. (2022) Secular Evolution of Continents and the Earth System. *Rev. Geophys.* 60.
- Champion D. C. and Smithies R. H. (2019) Geochemistry of Paleoarchean Granites of the East Pilbara Terrane, Pilbara Craton, Western Australia. In *Earth's Oldest Rocks* (eds. M. J. Van Kranendonk, V. C. Bennett, and J. E. Hoffmann). Elsevier B.V. pp. 487–518.
- Chowdhury P., Mulder J. A., Cawood P. A., Bhattacharjee S., Roy S., Wainwright A. N., Nebel O. and Mukherjee S. (2021) Magmatic thickening of crust in non-plate tectonic settings initiated the subaerial rise of Earth's first continents 3.3 to 3.2 billion years ago. *Proc. Natl. Acad. Sci.* 118, 1–8.
- Chowdhury P., Cawood P. A. and Mulder J. A. (2025) Subaerial Emergence of Continents on Archean Earth. *Annu. Rev. Earth Planet. Sci.*, 1–36.
- Gardiner N. J., Hickman A. H., Kirkland C. L., Lu Y., Johnson T. and Zhao J. X. (2017) Processes of crust formation in the early Earth imaged through Hf isotopes from the East Pilbara Terrane. *Precambrian Res.* 297, 56–76.
- Ge R.-F., Wilde S. A., Zhu W.-B. and Wang X.-L. (2023) Earth's early continental crust formed from wet and oxidizing arc magmas. *Nature* 623, 334–339.
- Hartnady, M.I.H., Johnson, T.E., Schorn, S., Hugh Smithies, R., Kirkland, C.L., Richardson, S.H., 2022. Fluid processes in the early Earth and the growth of continents. *Earth Planet. Sci. Lett.* 594, 117695.
- Hastie A. R., Law S., Bromiley G. D., Fitton J. G., Harley S. L. and Muir D. D. (2023) Deep formation of Earth's earliest continental crust consistent with subduction. *Nat. Geosci.*
- Hernández-Montenegro, J.D., Palin, R.M., Zuluaga, C.A., Hernández-Urbe, D., 2021. Archean continental crust formed by magma hybridization and voluminous partial melting. *Sci. Rep.* 11, 1–9. <https://doi.org/10.1038/s41598-021-84300-y>
- Hernández-Urbe, D., 2024. Generation of Archean oxidizing and wet magmas from mafic crustal overthickening. *Nat. Geosci.* 17, 809–813.
- Hickman A. H. (2021) East Pilbara Craton: a record of one billion years in the growth of Archean continental crust.
- Hickman A. H. (2023) Archean Evolution of the Pilbara Craton and Fortescue Basin. 1st ed., Springer International Publishing, Cham.
- Johnson T. E., Brown M., Gardiner N. J., Kirkland C. L. and Smithies R. H. (2017) Earth's first stable continents did not form by subduction. *Nature* 543, 239–242.
- Kendrick, J., Yakymchuk, C., 2020. Garnet fractionation, progressive melt loss and bulk composition variations in anatectic metabasites: Complications for interpreting the geodynamic significance of TTGs. *Geosci. Front.* 11, 745–763. <https://doi.org/10.1016/j.gsf.2019.12.001>
- Kendrick, J., Duguet, M., Yakymchuk, C., 2022. Diversification of Archean tonalite-trondhjemite-granodiorite suites in a mushy middle crust. *Geology* 50, 76–80. <https://doi.org/10.1130/G49287.1>
- Laurent, O., Björnsen, J., Wotzlaw, J.F., Bretscher, S., Pimenta Silva, M., Moyon, J.F., Ulmer, P., Bachmann, O., 2020. Earth's earliest granitoids are crystal-rich magma reservoirs tapped by silicic eruptions. *Nat. Geosci.* 13, 163–169. <https://doi.org/10.1038/s41561-019-0520-6>
- Law S., Hastie A. R., Young L. A. and Thordarson T. (2024) Formation of silicic crust on early Earth and young planetary bodies in an Iceland-like setting. *Commun. Earth Environ.* 5, 350.
- Law S. and Hastie A. R. (2025) Subduction Origin of the Nb Anomaly in Earth's Oldest Continents. *J. Petrol.* 66, egaf060.
- Law S., Hastie A. R., Young L. A. and Thordarson T. (2024) Formation of silicic crust on early Earth and young planetary bodies in an Iceland-like setting. *Commun. Earth Environ.* 5, 350.
- Lenardic, A., 2018. The diversity of tectonic modes and thoughts about transitions between them. *Philos. Trans. R. Soc. A Math. Phys. Eng. Sci.* 376
- Lourenço, D.L., Rozel, A.B., Ballmer, M.D., Tackley, P.J., 2020. Plutonic-Squishy Lid: a new global tectonic regime

generated by intrusive magmatism on Earth-like Planets. *Geochem. Geophys. Geosyst.* 21
<https://doi.org/10.1029/2019GC008756>.

Luffi P. and Ducea M. N. (2022) Chemical Mohometry: Assessing Crustal Thickness of Ancient Orogens Using Geochemical and Isotopic Data. *Rev. Geophys.* 60.

Martin, H., Moyen, J.F., Rapp, R., 2009. The sanukitoid series: Magmatism at the Archaean-Proterozoic transition. *Earth Environ. Sci. Trans. R. Soc. Edinburgh* 100, 15–33. <https://doi.org/10.1017/S1755691009016120>

Moyen, J.F., 2020. Archean granitoids: classification, petrology, geochemistry and origin. *Geol. Soc. Spec. Publ.* 489, 15–49.

Murphy M. E., Macdonald J. E., Fischer S., Gardiner N. J., White R. W. and Savage P. S. (2024) Silicon isotopes in an Archaean migmatite confirm seawater silicification of TTG sources. *Geochim. Cosmochim. Acta* 368, 34–49.

Nebel O., Vandenburg E. D., Capitanio F. A., Smithies R. H., Mulder J. and Cawood P. A. (2024) Early Earth “subduction”: short-lived, off-craton, shuffle tectonics, and no plate boundaries. *Precambrian Res.* 408, 107431.

Pourteau, A., Doucet, L.S., Blereau, E.R., Volante, S., Johnson, T.E., Collins, W.J., Li, Z.-X., Champion, D.C., 2020. TTG generation by fluid-fluxed crustal melting: Direct evidence from the Proterozoic Georgetown Inlier, NE Australia. *Earth Planet. Sci. Lett.* 550, 116548.

Rapp, R.P., Norman, M.D., Laporte, D., Yaxley, G.M., Martin, H., Foley, S.F., 2010. Continent formation in the archaean and chemical evolution of the cratonic lithosphere: Melt-rock reaction experiments at 3-4 GPa and petrogenesis of Archean Mg-diorites (Sanukitoids), *Journal of Petrology*. <https://doi.org/10.1093/petrology/egq017>

Smit, M.A., Musiyachenko, K.A., Goumans, J., 2024. Archaean continental crust formed from mafic cumulates. *Nat. Commun.* 15, 692. <https://doi.org/10.1038/s41467-024-44849-4>

Smithies, R.H., 2000. The Archaean tonalite-trondhjemite-granodiorite (TTG) series is not an analogue of Cenozoic adakite. *Earth Planet. Sci. Lett.* 182, 115–125. [https://doi.org/10.1016/S0012-821X\(00\)00236-3](https://doi.org/10.1016/S0012-821X(00)00236-3)

Smithies R. H., Champion D. C., Van Kranendonk M. J., Howard H. M. and Hickman A. H. (2005) Modern-style subduction processes in the Mesoarchaean: Geochemical evidence from the 3.12 Ga Whundo intra-oceanic arc. *Earth Planet. Sci. Lett.* 231, 221–237.

Tamblyn, R., Hermann, J., Hasterok, D., Sossi, P., Pettke, T., Chatterjee, S., 2023. Hydrated komatiites as a source of water for TTG formation in the Archean. *Earth Planet. Sci. Lett.* 603, 117982

Vandenburg E. D., Nebel O., Smithies R. H., Capitanio F. A., Miller L., Cawood P. A., Millet M.-A., Bruand E., Moyen J.-F., Wang X. and Nebel-Jacobsen Y. (2023) Spatial and temporal control of Archean tectonomagmatic regimes. *Earth-Science Rev.* 241, 104417.

Weinberg, R.F., Hasalová, P., 2015. Water-fluxed melting of the continental crust: a review. *Lithos* 212–215, 158–188.

Wiemer D., Schrank C. E., Murphy D. T., Wenham L. and Allen C. M. (2018) Earth’s oldest stable crust in the Pilbara Craton formed by cyclic gravitational overturns. *Nat. Geosci.* 11, 357–361.

Reviewer #3

(Remarks to the Author)

This study performed direct partial melting experiments on a 3.5 Ga metabasic source-rock from the Pilbara Craton to test the conditions for the formation of earliest felsic continental crust, which support a high-pressure (>45 km) origin as garnet stability field only at pressures ≥ 1.4 GPa. The results are promising and supportive of the conclusion, and can be accepted conditionally if my concerns listed below are thoroughly addressed.

In general, this study seems quite consistent (and similar) to another published study of the first author (Hastie et al., *Nature Geoscience*, 16, 816-821), which used a different starting material from a basaltic rock that of the southwestern Pacific Ontong Java Plateau; but for an unclear reason, this publication is not cited in this submission. I would suggest that the authors clearly emphasize the breakthrough of this submission compared to the previous one, if the starting compositions, the logic and design of experiments are quite similar.

Another concern is comparing the experimental results with the conflicting data from thermodynamic modeling (Johnson et al., 2017, *Nature* 543, 239-242), which used almost the same starting composition and target TTG for testing their results. A reasonable discussion may solve or release the puzzle of readers.

My third concern is that, accepting that a pressure of >1.4 GPa (>45 km) is needed for the formation of TTG rocks, how to conclude solely that a subduction-like tectonic environment has been the only possibility, i.e., how to exclude crustal drips/delamination model which is similarly consistent with such a high pressure (as the authors put in the abstract).

Version 1:

Reviewer comments:

Reviewer #2

[Editorial Note: Please also see attachments at end of file]

(Remarks to the Author)

This revised manuscript by Hastie et al. conducts high-pressure-temperature petrological experiments on a well-known lithology within the Archean igneous geochemistry community to understand the tectonic processes giving rise to Earth’s first continents. While the revised manuscript is an improvement over the original version dataset could be used to generate partition coefficients and inform future phase-equilibrium modelling, I do not believe it is suitable for publication in *Nature Communications*; the manuscript does not meet the journal’s threshold for conceptual advance and breadth of interest. The novelty is narrow. Despite the authors stating otherwise, this is not the first major and trace element experimental study of partial melting of an Archean amphibolite purported to be a potential source lithology for TTGs (see below). From the

perspective of a geochemist, I still believe it substantially overlaps with Hastie et al. (2023), Law et al. (2024) and Law and Hastie (2025), without a clear additional and substantial advance, beyond anything that will be evident to readers other than those from a niche portion of the experimental petrology community. Therefore, this contribution appears incremental for this venue and would be better suited to a specialist journal after substantial further development. The responses to my review comments don't adequately address my criticisms. Several core inferences still rest on evidence that is currently insufficient or internally inconsistent, and the broader geodynamic claims extend beyond what the new data can support. I hope that my comments and concerns help improve this manuscript.

As I mentioned previously, this manuscript doesn't strike me as particularly novel in using experimental petrology to investigate the origins of TTGs. For metabasites without oceanic plateau-like compositions, this isn't the first experimental study with major and trace elements measured in tandem (see e.g., Xiong et al., 2005). For experimental studies using Archean rocks (which inherently do not have modern oceanic plateau-like compositions), this is neither the first one to measure major elements (Rapp et al., 1991 and Rapp and Watson, 1995 use WR-40, a Neoproterozoic amphibolite from the Wyoming Craton), nor is it the first to measure major and trace elements in tandem (Adam et al., 2012 used Eoarchean Nuvvuagittuq amphibolites from the Superior Craton as their starting materials). These may be compositionally different from the Coonterunah metabasalts, but it does void the statement of priority.

One glaring issue that I don't believe I caught in the original manuscript is that the experiments in this study do not produce magmas with trondhjemitic compositions, which are typically associated with the high-pressure subgroup of TTG series magmas inferred to be produced by a plagioclase-free rutile and garnet-bearing source (e.g., Hoffmann et al., 2019; Laurent et al., 2024; Moyen and Martin, 2012; Moyen, 2020). This begs the question whether the results of the experiments can genuinely be considered representative of high-pressure TTGs, let alone those in the East Pilbara and thus calls into question the authors' inferences based on these results.

Another related issue involves the discrepancies between the experimental results and data from thermodynamic modelling (e.g., Johnson et al., 2017). I'm not satisfied with the authors' response to Reviewer 3's comment on this issue. The issues arising from the Magmatic NCKFMASHTOCr dataset of Holland et al. (2018) are unlikely to have a substantial impact on the stability of accessory phases. While the authors cite the corrigendum of this study as evidence for something being faulty with thermodynamic modelling, this dataset is entirely irrelevant to the results of Johnson et al. (2017) or White et al. (2017), as these studies used the metabasite NCKFMASHTO dataset of Green et al. (2016). Furthermore, the issues that the authors misattribute as being the main contributors to the discrepancy with their results have been fixed, so it is an easy (30 minutes tops) and worthwhile endeavour for the authors to compare their results with the most up-to-date thermodynamic modelling in MAGEMin or Thermocalc. It is unlikely that the thermodynamic datasets, which are constantly updated and informed by the results of numerous experiments, are incorrect, and I am not convinced of this on the basis of one experimental study.

The authors have removed their original argument towards the end of the main text about eclogite entrainment, but now jumps from inferences on TTGs made based on experimental results to unsubstantiated claims about the origin of the mafic protoliths employed in this study. This reads equally unsatisfactorily, given that this is unsubstantiated, beyond the scope of this study, and disregards substantial evidence to the contrary. Furthermore, on a related note, multiple points between lines 284-305 contradict each other with respect to inferences made about subduction based on TTGs.

I find some of the responses to my original comments to be unsatisfactory, particularly those relating to granitoids, geological observations from the Pilbara Craton, and those regarding geodynamics.

Granitoids:

TTGs can be IUGS-defined tonalites, trondhjemitic and granodiorites. However, the "TTG series", which I assume the authors are employing as the basis of their arguments, has a specific connotation in the Archean community (e.g., Moyen and Martin, 2012; Moyen, 2020; Spencer, 2025). Attributing all compositionally trondhjemitic, tonalitic and granodioritic compositions to the TTG series reflects a lack of engagement with the literature and the commonly used criteria that separate the TTG series from other Archean granitoids. Asserting that tonalites, trondhjemitic and granodiorites are compositionally identical to the TTG series as defined by seminal studies/articles such as Smithies and Champion (2000), Martin et al. (2005), Moyen and Martin (2012), Moyen (2020) and Laurent et al. (2024) without actively employing such criteria, perpetuates a misconception in the broader igneous community that the two are analogous.

The authors should perhaps consider the alternative models of Laurent et al. (2021) and Smit et al. (2024) as the processes outlined in this study have a direct impact on Nb contents, which is traditionally a proxy for residual rutile in TTGs.

Geological observations:

As I discussed in my comments on the previous round of reviews, granites and granodiorites are not the dominant silicic plutonic rock types for the Callina and Tambina Supersuites. While there are occurrences of granites in these supersuites as pointed out by the authors, and although I agree that some of these > 3.5-3.4 Ga granites are likely produced by low-degree melting of amphibolites, in reality, these are volumetrically subordinate. To use the example that the authors provide as evidence, the North Pole Monzogranite is a relatively small intrusion. Likewise, rhyolites are not particularly germane to the argument as many of these are not genetically related to TTGs and do not share the same source (e.g., Smithies et al., 2019), with some exceptions, including the Coucal "TTG flow" of Smithies et al. (2009).

In their rebuttal, the authors state, "one needs to estimate proportions based on available geochemical data." I did just that in the original round of reviews, providing two figures using two methods to demonstrate that granites and granodiorites are subordinate to tonalites and trondhjemitites in the Callina and Tambina Supersuites. I have reattached the two figures from the original round of reviews and recommend that the authors reread my original comments, as their response has been unsatisfactory.

I'm familiar with Kemp et al. (2023), and it is certainly an important study. However, it doesn't exist in a vacuum, especially with respect to Hf isotope data in granitoids and has its shortfalls. For one, Kemp et al. (2023) do not consider the work of Gardiner et al. (2017, 2021). Additionally, samples from all terranes are presented together in the ϵHf -Age diagrams rather than the samples from each terrane separately.

Because the different terranes of the Pilbara evolved as independent entities, this can lead to some spurious inferences when grouped on the same diagrams. I believe this could explain the authors' incorrect assertion that "younger granodiorites forming by remelting the tonalites" is not supported by isotope data, at least to 3.2 Ga" in response to my comment about the presence of reworked components in granitoids from some of the 3.32-3.29 Ga Emu Pool Supersuite intrusions and many of the 3.27-3.22 Ga Cleland Supersuite intrusions. To demonstrate the presence of this component from 3.3 Ga onwards, I've plotted all available Hf isotope in zircon data for magmatic grains (within $\pm < 5\%$ discordance) from East Pilbara Terrane granitoids versus time in Fig. 3. As can be seen, both sample mean ϵHf and individual zircon grain ϵHf begins to decrease around 3.3 Ga (coinciding with the Emu Pool Supersuite) and gets especially pronounced and subchondritic around 3.27 Ga (coinciding with the Cleland Supersuite). This shift is best reconciled with the presence of a reworked component. This isn't to say that all granitoids from these supersuites contain the component, as there are clearly intrusions with suprachondritic ϵHf , but many do.

Other lines of evidence exist in addition to the changes in granitoid geochemistry towards more potassic compositions (Vandenburg et al., 2023) and the outlined Hf isotope systematics. Gardiner et al. (2021) also provide further evidence for reworked components based on the presence of xenocrysts in the central portions of Emu Pool Supersuite intrusions, suggesting inheritance occurred at depth in their sources, rather than due to wall rock entrainment. Furthermore, oxygen isotope data presented by Smithies et al. (2021) and Johnson et al. (2022) from granitoids of the Emu Pool and Cleland Supersuites are best reconciled with the presence of a reworked component. Altogether, evidence from the literature most parsimoniously suggests that the latter two Paleoproterozoic granitoid supersuites of the East Pilbara increasingly incorporated a reworked component, as opposed to being produced by low-degree melting of an amphibolite source. Therefore, the results of this study are not entirely relevant to the <3.3 Ga granitoids of the Pilbara and the authors should be more explicit to state that these results apply mostly to the > 3.3 Ga supersuites.

I understand that this is not a review paper, but the authors need to take a more holistic approach when making inferences based on their results and account for the existing literature from the Pilbara. The four Paleoproterozoic granitoid supersuites of the East Pilbara Terrane were emplaced contemporaneously with the ultramafic-felsic lavas of the Pilbara Supergroup. If subduction-like processes produced the TTGs, then one would expect that the Pilbara Supergroup lavas would bear geochemical signatures consistent with such a tectonomagmatic setting, given their close spatio-temporal association. However, there is no evidence for such an origin for the lavas of the Pilbara Supergroup, and they are better explained by variably crustally contaminated volcanism in a mafic plateau setting above a mantle upwelling (e.g., Brown et al., 2024; Hasenstab et al., 2021; Maier et al., 2009; Murphy et al., 2021; Nebel et al., 2014; Puchtel et al., 2022; Smithies et al., 2005; Tympel et al., 2021). Thus, as concluded by numerous studies using field, geochronological, structural, and geochemical data, geological context of the EPT in the Paleoproterozoic remains most parsimoniously explained by processes not diagnostic of plate tectonics (e.g., Brown et al., 2024; Campbell and Davies, 2017; Gardiner et al., 2017; Hasenstab et al., 2021; Hawkesworth and Kemp, 2021; Hickman, 2021, 2023; Johnson et al., 2017, 2022; Roberts et al., 2022; Smithies et al., 2005a, 2009, 2021; Van Kranendonk et al., 2015, 2019; Wiemer et al., 2018). While experiments can offer interpretations and possible explanations of data, experimental results do not constitute proof of the operation of a given process.

Geodynamics:

The planetary angle and broad geodynamic statements are not anchored by new data in this study and still read as overreach; the authors' response to my original comment on this matter does not rectify this issue.

As I said in the original round of reviews, plate tectonics is a very specific type of geodynamic regime. To quote Stern and Gerya (2018), plate tectonics is defined as "a theory of global tectonics powered by subduction in which the lithosphere is divided into a mosaic of strong lithospheric plates, which move on and sink into weaker ductile asthenosphere. Three types

of localized plate boundaries form the interconnected global network: new oceanic plate material is created by seafloor spreading at mid-ocean ridges, old oceanic lithosphere sinks at subduction zones, and two plates slide past each other along transform faults. The negative buoyancy of old dense oceanic lithosphere, which sinks in subduction zones, mostly powers plate movements.”

The return of mafic crust to the mantle (either through dripping or subduction) would, of course, be accompanied by rifting elsewhere to replace the volume of crust returned to the mantle, but it need not comprise a global, interlinked network of ridges, transform faults and subduction zones. As shown by multiple numerical modelling studies (e.g., Capitanio et al., 2019, 2020, 2022; Lenardic, 2018; Lourenço et al., 2020), this does not necessarily imply plate tectonics. If it did, then Venus would also be considered to have a “proto-type of plate tectonics” (e.g., Byrne et al., 2021; Capitanio et al., 2024; Gillmann et al., 2025), thereby voiding part of this pitch aimed at a more general scientific audience.

The tectonic regime of Earth has evolved along a continuum through its history (see e.g., the review in Cawood et al., 2022), so by the authors’ overly broad definition, every possible geodynamic regime with lithospheric recycling would be a “proto-type of plate tectonics.”

Therefore, I think the wording here needs to change, as the findings of this study do not implicate plate tectonics, even in a subduction-like scenario.

Line-by-line comments:

Line 34: What’s meant by “continental” here? Does this refer to the UCC, MCC, LCC or the bulk CC? These all mean different things and have different implications.

Line 42: Absence? They’re certainly rare, but Hadean xenocrystic and detrital zircons have been found in multiple cratons (see e.g., Condie, 2019 for a brief review).

Fig. 1: If the authors refer to the Callina and Tambina Supersuites in Lines 85-90, then it would make sense to demarcate the distribution of these supersuites on the map, especially when the Split Rock Supersuite is already on there. Also, I’m confused as to why the Fortescue Group and Kurrana Terrane are included on this map, while the Central Pilbara Tectonic Zone is not, as the former two are also not part of the East Pilbara Terrane. I’d also recommend symbolizing the sample locality on the map as a dot, perhaps.

Line 100: For metabasites without oceanic plateau-like compositions, this isn’t the first experimental study with major and trace elements measured in tandem (see e.g., Xiong et al., 2005). For experimental studies using Archean rocks (which inherently do not have modern oceanic plateau-like compositions), this is neither the first one to measure major elements (Rapp et al., 1991 and Rapp and Watson, 1995 use WR-40, an Archean amphibolite from the Wyoming Craton), nor is it the first to measure major and trace elements in tandem (Adam et al., 2012 used Nuvvuagittuq amphibolites as their starting materials). These may be compositionally different from the Coonterunah metabasalts, but it does void the statement of priority.

Line 110-111: Sample 179789 does, in fact, show evidence for crustal contamination by older silicic crust, as demonstrated in the SI of Johnson et al. (2017), so stating that the Coonterunah metabasalts “show no evidence that their parental magmas have been contaminated with older silicic crust” is inaccurate.

Lines 167-168: See discussion above about granitoids and geological observations.

Fig. 3: Given the above comments, I recommend the authors only plot > 3.32 Ga granitoids from the East Pilbara as shaded areas on their primitive mantle normalized extended trace element diagrams. Including the younger granitoids will invariably lead to spurious comparisons.

Lines 214-215: Again, see the discussion above about granitoids and geological observations. Each of the granitoid complexes in the East Pilbara is comprised of sets of composite intrusions spanning > 700 Myr of Earth’s history and if one looks at a geological map of the area, it is evident that the < 3.32 Ga intrusions are more widespread. I believe this is a misinterpretation once again of Hickman (2004), where the line about granitoids being “dominantly monzogranite and granodiorite” is in reference to the granitoids of the Pilbara Craton as a whole, rather than the EPT, let alone the Callina and Tambina Supersuites. As I have discussed above, the authors’ assertion that granites and granodiorites are “volumetrically dominant in the East Pilbara plutonic centres” is not germane to the central comparison, given that this volumetric dominance is the consequence of later magmatism incorporating an increasing amount of reworked components, rather than due to conditions during the generation of TTGs of the Callina and Tambina Supersuites.

Lines 284-288: Not all TTGs are derived from eclogites, some are derived from amphibolites, as demonstrated by field evidence (e.g., Kendrick et al., 2024; Pourteau et al., 2020). Ascribing the entire source to eclogites is inaccurate, especially given that high-pressure group TTGs are not as common as low-pressure group TTGs in the Pilbara (e.g., Champion and Smithies, 2019; Vandenburg et al., 2023).

Line 289: This study does not show that a wet and deep tectonic environment is required to produce the Nb anomalies in the F2 Coonterunah basalts. This is granitoid petrology paper, not one on the mantle source of CF2. For what it's worth crustal contamination better explains many of the Nb anomalies in the CF2 basalts (e.g., Brown et al., 2024; Johnson et al., 2017). Crustal contamination can be quite cryptic in many cases as well (see e.g., Vite-Sánchez et al., 2024).

Lines 296-298: In my opinion, this is overly speculative, lacks proper substantiation and diverts from the core message of this study.

Lines 302-303: If subduction is occurring at a flat angle, then doesn't that contradict the statement in line 298 about the existence of steep subduction with a mantle wedge?

Lines 304-305: Fractional crystallization does not remove the mantle signature in modern adakites, even on the rhyolitic side of things, so I don't see how that would be the case in the Archean with TTGs. If there is subduction, it is unlikely for there to be a mantle wedge, not only because of the aforementioned considerations (see also Smithies, 2000; Martin et al., 2005), but also because all that armouring required to prevent mantle signatures in the TTGs would ultimately have to melt, producing sanukitoids and metasomatized mantle-derived mafic lavas, which are conspicuously absent from the East Pilbara Terrane.

Lines 306-311: Wrt the ability of subduction-like processes to emit gases into the atmosphere, the emplacement style of the volcanoes is a critical question. The geological evidence is overwhelmingly in favour of subaqueous emplacement (e.g., Flament et al., 2008; Hickman, 2023) in most places, such that the hydrostatic pressure of the overlying oceanic water column affected the nature of emitted gases (e.g., Gaillard et al., 2011). This is consistent with multiple lines of evidence for very low sulfate concentrations in the Archean ocean. There are many more related pieces of evidence regarding Archean atmosphere-hydrosphere-lithosphere interaction that should be considered if trying to pursue this route (e.g., Kamber and Ossa-Ossa, 2025).

References

- Adam J., Rushmer T., O'Neil J. and Francis D. (2012) Hadean greenstones from the Nuvvuagittuq fold belt and the origin of the Earth's early continental crust. *Geology* 40, 363–366.
- Brown M., Pearce J. A. and Johnson T. E. (2024) Is plate tectonics a post-Archean phenomenon? A petrological perspective. *J. Geol. Soc. London*. 181.
- Byrne P. K., Ghail R. C., Şengör A. M. C., James P. B., Klimczak C. and Solomon S. C. (2021) A globally fragmented and mobile lithosphere on Venus. *Proc. Natl. Acad. Sci.* 118.
- Campbell I. H. and Davies D. R. (2017) Raising the continental crust. *Earth Planet. Sci. Lett.* 460, 112–122.
- Capitanio F. A., Nebel O., Cawood P. A., Weinberg R. F. and Clos F. (2019) Lithosphere differentiation in the early Earth controls Archean tectonics. *Earth Planet. Sci. Lett.* 525, 115755.
- Capitanio F. A., Nebel O. and Cawood P. A. (2020) Thermochemical lithosphere differentiation and the origin of cratonic mantle. *Nature* 588, 89–94.
- Capitanio F. A., Nebel O., Moya J. and Cawood P. A. (2022) Craton Formation in Early Earth Mantle Convection Regimes. *J. Geophys. Res. Solid Earth* 127.
- Capitanio F. A., Kerr M., Stegman D. R. and Smrekar S. E. (2024) Ishtar Terra highlands on Venus raised by craton-like formation mechanisms. *Nat. Geosci.* 17, 740–746.
- Cawood P. A., Chowdhury P., Mulder J. A., Hawkesworth C. J., Capitanio F. A., Gunawardana P. M. and Nebel O. (2022) Secular Evolution of Continents and the Earth System. *Rev. Geophys.* 60.
- Condie K. C. (2019) Earth's Oldest Rocks and Minerals. In *Earth's Oldest Rocks* (eds. M. J. Van Kranendonk, V. C. Bennett, and J. E. Hoffmann). Elsevier B.V. pp. 239–253.
- Flament, N., Coltice, N. and Rey, P.F., 2008. A case for late-Archean continental emergence from thermal evolution models and hypsometry. *Earth and Planetary Science Letters*, 275(3-4), pp.326-336.
- Gaillard, F., Scaillet, B. and Arndt, N.T., 2011. Atmospheric oxygenation caused by a change in volcanic degassing pressure. *Nature*, 478(7368), pp.229-232.
- Gardiner N. J., Hickman A. H., Kirkland C. L., Lu Y., Johnson T. and Zhao J. X. (2017) Processes of crust formation in the early Earth imaged through Hf isotopes from the East Pilbara Terrane. *Precambrian Res.* 297, 56–76.

- Gardiner N. J., Mulder J. A., Nebel O., Kirkland C. L. and Johnson T. E. (2021) Palaeoarchaeon TTGs of the Pilbara and Kaapvaal cratons compared; an early Vaalbara supercraton evaluated. *South African J. Geol.* 124, 1–16.
- Gillmann C., Arney G. N., Avicé G., Dyar M. D., Golabek G. J., Gülcher A. J. P., Johnson N. M., Lefèvre M. and Widemann T. (2025) Venus. In *Treatise on Geochemistry* (eds. A. D. Anbar and D. Weis). Elsevier. pp. 289–323.
- Green E. C. R., White R. W., Diener J. F. A., Powell R., Holland T. J. B. and Palin R. M. (2016) Activity–composition relations for the calculation of partial melting equilibria in metabasic rocks. *J. Metamorph. Geol.* 34, 845–869.
- Hastie A. R., Law S., Bromiley G. D., Fitton J. G., Harley S. L. and Muir D. D. (2023) Deep formation of Earth's earliest continental crust consistent with subduction. *Nat. Geosci.*
- Hasenstab E., Tusch J., Schnabel C., Marien C. S., Van Kranendonk M. J., Smithies H., Howard H., Maier W. D. and Münker C. (2021) Evolution of the early to late Archean mantle from Hf-Nd-Ce isotope systematics in basalts and komatiites from the Pilbara Craton. *Earth Planet. Sci. Lett.* 553, 116627.
- Hawkesworth C. and Kemp T. (2021) A Pilbara perspective on the generation of Archean continental crust. *Chem. Geol.* 578, 120326.
- Hickman A. H. (2021) East Pilbara Craton: a record of one billion years in the growth of Archean continental crust.,
- Hickman A. H. (2023) *Archean Evolution of the Pilbara Craton and Fortescue Basin*. 1st ed., Springer International Publishing, Cham.
- Hoffmann J. E., Zhang C., Moyén J.-F. and Nagel T. J. (2019) The Formation of Tonalites–Trondjemite–Granodiorites in Early Continental Crust. In *Earth's Oldest Rocks* (eds. M. J. Van Kranendonk, V. C. Bennett, and J. E. Hoffmann). Elsevier B.V. pp. 133–168.
- Holland T. J. B., Green E. C. R. and Powell R. (2018) Melting of Peridotites through to Granites: A Simple Thermodynamic Model in the System KNCFMASHTOCr. *J. Petrol.* 59, 881–900.
- Johnson T. E., Brown M., Gardiner N. J., Kirkland C. L. and Smithies R. H. (2017) Earth's first stable continents did not form by subduction. *Nature* 543, 239–242.
- Johnson T. E., Kirkland C. L., Lu Y., Smithies R. H., Brown M. and Hartnady M. I. H. (2022) Giant impacts and the origin and evolution of continents. *Nature* 608, 330–335.
- Kamber B. S. and Ossa Ossa F. (2025) Evolution of continental crust and sedimentary rock chemistry through time. In *Treatise on Geochemistry* (eds. A. D. Anbar and D. Weis). Elsevier. pp. 729–773.
- Kendrick J., Duguet M., Kirkland C. L., Liebmann J., Moser D. E., Vervoort J. D. and Yakymchuk C. (2024) Testing the TTG–Metabasite Connection in the Southern Superior Province: an Integrated Geochemical, Isotopic, and Petrogenetic Modelling Approach. *J. Petrol.* 65.
- Laurent O., Björnsen J., Wotzlaw J. F., Bretscher S., Pimenta Silva M., Moyén J. F., Ulmer P. and Bachmann O. (2020) Earth's earliest granitoids are crystal-rich magma reservoirs tapped by silicic eruptions. *Nat. Geosci.* 13, 163–169.
- Laurent O., Guitreau M., Bruand E. and Moyén J. F. (2024) At the Dawn of Continents: Archean Tonalite-Trondjemite-Granodiorite Suites. *Elements* 20, 174–179.
- Law S., Hastie A. R., Young L. A. and Thordarson T. (2024) Formation of silicic crust on early Earth and young planetary bodies in an Iceland-like setting. *Commun. Earth Environ.* 5, 350.
- Law S. and Hastie A. R. (2025) Subduction Origin of the Nb Anomaly in Earth's Oldest Continents. *J. Petrol.* 66, egaf060.
- Law S., Hastie A. R., Young L. A. and Thordarson T. (2024) Formation of silicic crust on early Earth and young planetary bodies in an Iceland-like setting. *Commun. Earth Environ.* 5, 350.
- Lenardic, A., 2018. The diversity of tectonic modes and thoughts about transitions between them. *Philos. Trans. R. Soc. A Math. Phys. Eng. Sci.* 376
- Lourenço D. L., Rozel A. B., Ballmer M. D. and Tackley P. J. (2020) Plutonic-Squishy Lid: A New Global Tectonic Regime Generated by Intrusive Magmatism on Earth-Like Planets. *Geochemistry, Geophys. Geosystems* 21.
- Maier W. D., Barnes Stephen J., Campbell I. H., Fiorentini M. L., Peltonen P., Barnes Sarah Jane and Smithies R. H. (2009) Progressive mixing of meteoritic veneer into the early Earths deep mantle. *Nature* 460, 620–623.
- Martin H., Smithies R. H., Rapp R., Moyén J.-F. and Champion D. (2005) An overview of adakite, tonalite–trondjemite–granodiorite (TTG), and sanukitoid: relationships and some implications for crustal evolution. *Lithos* 79, 1–24.

- Moyen J.-F. (2020) Archean granitoids: classification, petrology, geochemistry and origin. *Geol. Soc. London, Spec. Publ.* 489, 15–49.
- Moyen J. F. and Martin H. (2012) Forty years of TTG research. *Lithos* 148, 312–336.
- Murphy D., Rizo H., O’Neil J., Hepple R., Wiemer D., Kemp A. and Vervoort J. (2021) Combined Sm-Nd, Lu-Hf, and ¹⁴²Nd study of Paleoproterozoic basalts from the East Pilbara Terrane, Western Australia. *Chem. Geol.* 578, 120301.
- Nebel O., Campbell I. H., Sossi P. A. and Van Kranendonk M. J. (2014) Hafnium and iron isotopes in early Archean komatiites record a plume-driven convection cycle in the Hadean Earth. *Earth Planet. Sci. Lett.* 397, 111–120.
- Pourteau A., Doucet L. S., Blereau E. R., Volante S., Johnson T. E., Collins W. J., Li Z.-X. and Champion D. C. (2020) TTG generation by fluid-fluxed crustal melting: Direct evidence from the Proterozoic Georgetown Inlier, NE Australia. *Earth Planet. Sci. Lett.* 550, 116548.
- Puchtel I. S., Nicklas R. W., Slagle J., Horan M., Walker R. J., Nisbet E. G. and Locmelis M. (2022) Early global mantle chemical and isotope heterogeneity revealed by the komatiite-basalt record: The Western Australia connection. *Geochim. Cosmochim. Acta* 320, 238–278.
- Rapp R. P. and Watson E. B. (1995) Dehydration Melting of Metabasalt at 8–32 kbar: Implications for Continental Growth and Crust-Mantle Recycling. *J. Petrol.* 36, 891–931.
- Rapp R. P., Watson E. B. and Miller C. F. (1991) Partial melting of amphibolite/eclogite and the origin of Archean trondhjemites and tonalites. *Precambrian Res.* 51, 1–25.
- Roberts N. M., Tikoff B. and Salerno R. A. (2022) Greenstone-Up Shear Sense at the Margin of the Mt Edgar Dome, East Pilbara Terrane: Implications for Dome and Keel Formation in the Early Earth. *Tectonics* 41, 1–20.
- Smit M. A., Musiyachenko K. A. and Goumans J. (2024) Archean continental crust formed from mafic cumulates. *Nat. Commun.* 15, 692.
- Smithies R. H. (2000) The Archean tonalite-trondhjemite-granodiorite (TTG) series is not an analogue of Cenozoic adakite. *Earth Planet. Sci. Lett.* 182, 115–125.
- Smithies R. H. and Champion D. C. (2000) The Archean high-Mg diorite suite: Links to Tonalite-Trondhjemite-Granodiorite magmatism and implications for early Archean crustal growth. *J. Petrol.* 41, 1653–1671.
- Smithies R. H., Van Kranendonk M. J. and Champion D. C. (2005) It started with a plume - Early Archean basaltic proto-continental crust. *Earth Planet. Sci. Lett.* 238, 284–297.
- Smithies R. H., Champion D. C. and Van Kranendonk M. J. (2009) Formation of Paleoproterozoic continental crust through infracrustal melting of enriched basalt. *Earth Planet. Sci. Lett.* 281, 298–306.
- Smithies R. H., Champion D. C. and Van Kranendonk M. J. (2019) The Oldest Well-Preserved Felsic Volcanic Rocks on Earth: Geochemical Clues to the Early Evolution of the Pilbara Supergroup and Implications for the Growth of a Paleoproterozoic Protocontinent. In *Earth’s Oldest Rocks* (eds. M. J. Van Kranendonk, V. C. Bennett, and J. E. Hoffmann). Elsevier. pp. 463–486.
- Smithies R. H., Lu Y., Kirkland C. L., Johnson T. E., Mole D. R., Champion D. C., Martin L., Jeon H., Wingate M. T. D. and Johnson S. P. (2021) Oxygen isotopes trace the origins of Earth’s earliest continental crust. *Nature* 592, 70–75.
- Spencer L. M. (2025) *Insights into Sanukitoid Formation and Evolution from Novel Stable Isotope Systems*. Cardiff University.
- Stern R. J. and Gerya T. (2018) Subduction initiation in nature and models: A review. *Tectonophysics* 746, 173–198.
- Tympel J. F., Hergt J. M., Maas R., Woodhead J. D., Greig A., Bolhar R. and Powell R. (2021) Mantle-like Hf–Nd isotope signatures in ~3.5 Ga greenstones: No evidence for Hadean crust beneath the East Pilbara Craton. *Chem. Geol.* 576, 120273.
- Vite-Sánchez O., Ross P.-S. and Mercier-Langevin P. (2024) Mafic to intermediate volcanic rocks of the Blake River Group, Abitibi greenstone belt, Canada: Geochemistry, petrogenesis and relation with VMS deposits. *Precambrian Res.* 404, 107331.
- Van Kranendonk M. J., Hugh Smithies R., Griffin W. L., Huston D. L., Hickman A. H., Champion D. C., Anhaeusser C. R. and Pirajno F. (2015) Making it thick: A volcanic plateau origin of Palaeoproterozoic continental lithosphere of the Pilbara and Kaapvaal cratons. *Geol. Soc. Spec. Publ.* 389, 83–111.

Van Kranendonk M. J., Smithies R. H. and Champion D. C. (2019) Paleoproterozoic Development of a Continental Nucleus: The East Pilbara Terrane of the Pilbara Craton, Western Australia. In *Earth's Oldest Rocks* (eds. M. J. Van Kranendonk, V. C. Bennett, and J. E. Hoffmann). Elsevier B.V. pp. 437–462.

White R. W., Palin R. M. and Green E. C. R. (2017) High-grade metamorphism and partial melting in Archean composite grey gneiss complexes. *J. Metamorph. Geol.* 35, 181–195.

Xiong X. L., Adam J. and Green T. H. (2005) Rutile stability and rutile/melt HFSE partitioning during partial melting of hydrous basalt: Implications for TTG genesis. *Chem. Geol.* 218, 339–359.

Reviewer #3

(Remarks to the Author)

I found that, in the response and revised ms, all my concerns have been adequately addressed, which makes me suggest acceptance. Congratulations to the authors for a nice work that shows the power of experimental study on geodynamics.

Version 2:

Reviewer comments:

Reviewer #4

(Remarks to the Author)

Hastie et al. present high-pressure melting experimental data from a Paleoproterozoic basalt (Coonterunah Subgroup) in the East Pilbara Terrane to investigate the melting conditions and geodynamic context of early Archean continental crust.

I am not an expert in high-pressure melting experiments and am unable to comment on the quality of the experimental design or results. The handling editor requested that I assess the novelty of this study, the presentation of Pilbara Craton geology, and the discussion of the geodynamic implications.

I am satisfied that this study is suitably novel and impactful to be published in *Nature Communications*. There have been comparable studies specifically investigating melt generation from the Coonterunah Subgroup basalts, but these are based on thermodynamic modelling. The new experimental data presented by Hastie et al. is a valuable addition to the literature.

I have two main suggestions for improvement:

(1) A clearer definition of the specific time-frame the study is relevant to.

The title states 'Early Archean', the abstract mentions 4.3–3.5 Ga and 4.0–3.5 Ga time periods, lines 83–93 appear to indicate the study is specifically focusing on generation of the Callina and Tambina supersuites (~3.5–3.4 Ga), and several places in the text mention Eoarchean-Paleoproterozoic or Hadean-Paleoproterozoic crustal formation.

Do the experimental results only pertain to the generation of Paleoproterozoic TTGs in the East Pilbara Terrane? If so, this could be stated more explicitly. Note that the Eoarchean record of the East Pilbara Terrane is sparse so line 77 needs amending (the Paleoproterozoic might be extensive, but the Eoarchean is only known from rare enclaves and detrital zircon and apatite grains). It is possible that the composition and crust-forming processes of the Eoarchean crust of the East Pilbara Terrane differ from the preserved Paleoproterozoic crust (compare Hickman, 2023, *Archean Evolution of the Pilbara Craton and Fortescue Basin* and Kharkongor et al. 2025, *Geology*).

This point is also true for other Archean terranes where any Hadean-Eoarchean record is primarily preserved as detrital or xenocrystic zircon grains. There is no consensus on the composition of the original host rocks of Hadean-Eoarchean zircons (a quick survey of recent literature on Hadean zircon trace element chemistry includes estimates ranging from peraluminous granites, andesites, mafic rocks, and TTGs). As written the paper implies that the Paleoproterozoic TTGs of the EPT are representative of all early continental crust (including Eoarchean-Hadean). I think this is a reasonable assertion but the debate regarding the composition and origin of Eoarchean-Hadean crust and the possibility that it differed from the more widely preserved Paleo-Mesoproterozoic crust should be acknowledged.

(2) The discussion of lithospheric recycling on the early Earth could be framed in the context of recent geodynamic modelling studies.

Inferring Early Earth geodynamic settings from geochemical data is highly contested and non-unique. I recognise that the authors have already made a significant effort to tone-down this aspect of the manuscript during earlier reviews. However, the inferences of the geodynamic setting of early Archean crust formation could be strengthened by discussing this aspect of the study in the context of recent geodynamic modelling.

The study advocates for deep subduction and 'proto' or 'primitive' plate tectonics to generate Paleoproterozoic TTGs in the East Pilbara Terrane. 'Proto plate tectonics' and 'primitive plate tectonics' are nebulous terms and there have been recent efforts to clearly define the difference between modern plate tectonics and the geodynamic environments operating on the early Earth (see Cawood et al. 2022 for an accessible summary). Numerical and thermomechanical modelling also highlight

important differences in geodynamic settings on a hotter early Earth compared to modern plate tectonics. These studies recognise several possible geodynamic modes on early Earth, for example, stagnant lid, squishy lid, mobile lid modes.

Subduction or lithospheric dripping are cited as possible mechanisms for generating high-pressure TTG melts but the authors lean in favour of subduction and a plate tectonic like regime (e.g., in the title and abstract). Both these processes are consistent with 'squishy' or 'mobile' lid tectonic modes. Many readers may find these definitions more palatable than referring explicitly to plate tectonics. In my opinion, using experimental data to demonstrate a diversity of melt-generating environments on the early Earth is an important and impactful finding alone. Describing the geodynamic setting of melting might be an issue of semantics, but at least using the more recent definitions outlined above provides a common framework for communicating the geodynamic setting the authors are envisaging.

Relevant geodynamic modelling literature:

Capitanio, F. A., Nebel, O., & Cawood, P. A. (2020). Thermochemical lithosphere differentiation and the origin of cratonic mantle. *Nature*, 588(7836), 89–94. <https://doi.org/10.1038/s41586-020-2976-3>

Capitanio, F. A., Nebel, O., Cawood, P. A., Weinberg, R. F., & Chowdhury, P. (2019). Reconciling thermal regimes and tectonics of the early Earth. *Geology*, 47(10), 923–927. <https://doi.org/10.1130/g46239.1>

Cawood, P. A., Chowdhury, P., Mulder, J. A., Hawkesworth, C. J., Capitanio, F. A., Gunawardana, P. M., & Nebel, O. (2022). Secular evolution of continents and the Earth system. *Reviews of Geophysics*, 60, e2022RG000789. <https://doi.org/10.1029/2022RG000789>

Rozel, A. B., Golabek, G. J., Jain, C., Tackley, P. J., & Gerya, T. (2017). Continental crust formation on early Earth controlled by intrusive magmatism. *Nature*, 545(7654), 332–335. <https://doi.org/10.1038/nature22042>

Minor points:

Inconsistent naming of the study area. The study area is referred to as 'Pilbara terrane', 'northern Pilbara craton', 'East Pilbara'. The study focuses on the EAST PILBARA TERRANE, which is a terrane within the Pilbara Craton. This should be updated throughout the text.

Use of 'granitic'. There are few true granites in the Pilbara (i.e., Kfsp-rich rocks). 'Granitoids' would be a more accurate description for the felsic plutonic rocks in the East Pilbara Terrane.

Version 3:

Reviewer comments:

Reviewer #4

(Remarks to the Author)

I am satisfied that the authors have addressed my queries.

Response to reviewers for manuscript NCOMMS-25-59996-T

Reviewer #1 (Remarks to the Author):

General comments

This is a great and timely study addressing the origin of Paleoproterozoic continental crust in East Pilbara, northwestern Australia. The study addresses first order, fundamental questions in Earth science related to the origin and tectonic setting of early continental crust. The authors use major and trace element geochemical data to constrain the formation depth of granitic melts that were generated by well-constrained experiments. The authors provide a good introduction to the study topic. Objectives are clear. Arguments presented in the manuscript agree with the data reported in the manuscript and the data in the literature. The results of the study show that most of the assumptions made on the geology of East Pilbara by many previous studies are not supported by robust experimental data. So, these assumptions need to be revised. The topic of the manuscript is of great interest to the readers of Nature Communications. I strongly recommend the publication of the study as it is. It will be a well-cited paper. After the publication of this paper, geologists will look at the origin of Archean continental crust in East Pilbara from a totally new perspective. The East Pilbara Archean terrain represents one of the key regions that is used to test Archean tectonic models. Both plate tectonic and non-plate tectonic models are proposed to explain the rock record in East Pilbara. Because of its dome and basin structure, the East Pilbara has been at the center of geological debate on the origin of Archean continental crust for many decades.

We thank the reviewer for the highly supportive comments.

Specific comments

Line 1: I recommend replacement of “early Archean” with “Paleoproterozoic”. This is up to the authors.

Wording replaced.

Lines 35-37: I am not aware of any field data providing evidence for “drips/delamination”.

Re-worded.

Line 44: Add “crust” after “continental”.

‘crust’ added

References need to be edited.

Done

Reviewer #2 (Remarks to the Author):

This study by Hastie et al. conducts high-pressure-temperature petrological experiments on a well-known lithology within the Archean igneous geochemistry community in an attempt to understand the tectonic processes giving rise to Earth's first continents. While the dataset could be used to generate partition coefficients and inform future phase-equilibrium modelling, I cannot recommend it for publication in Nature Communications; the manuscript does not meet the journal's threshold for conceptual advance and breadth of interest. The novelty is narrow because it substantially overlaps with Hastie et al. (2023), Law et al. (2024) and Law and Hastie (2025), without a clear additional and substantial advance.

We are afraid that there is some confusion over the previous studies by Hastie and Law. The latter investigate modern oceanic plateau source regions as analogues of the early Earth's mafic crustal surface. The study here is, to our knowledge, the **first** and **only** major and trace element experimental study of an ancient metabasic source region from Pilbara that **does not** have modern oceanic plateau compositions. The two starting compositions (the ones from Hastie et al. – 1187-8 and 1187-10) and this new study (EPT) are **completely and utterly different with regards to both major and trace element concentrations**:

Sample	SiO ₂	TiO ₂	Al ₂ O ₃	FeO(t)	MnO	MgO	CaO	Na ₂ O	K ₂ O	P ₂ O ₅
EPT2019	48.069	2.17	14.855	13.75	0.24	7.72	9.68	2.42	0.96	0.14
EPT2021	47.822	2.14	14.639	14.44	0.24	7.66	9.59	2.38	0.94	0.14
EPT2024	48.10	2.07	15.58	13.84	0.25	7.59	9.17	2.32	0.95	0.13
1187-8	49.22	0.75	14.87	9.76	0.17	9.89	12.39	1.64	0.09	0.06
1187-10	49.89	0.73	14.53	9.85	0.16	10.20	12.77	1.73	0.10	0.06

Of the ten major elements, 8 are completely different in the new starting mix (highlighted green) and the trace element concentrations are an order of magnitude different. **We take full responsibility for this confusion and now modify the text to make it clear that the previous studies are not related at all (Lines 97-105).** However, Pilbara is at the **height** of controversy regarding the evolution of the early continental crust so the results of this study will be of great interest to the scientific community and will be highly sort after.

Major comments:

The cornerstone of the ≥ 2.0 GPa requirement is the appearance of rutile. In the draft, rutile is “sub-micron,” identified via acicular habit and a TiO_2 spike on EDS, and explicitly stated to be too small to analyze. No Raman, micro-XRD, or TEM confirmation is provided, and no modal estimates are reported. Given that the Nb argument hinges on rutile, this evidence is not adequate. The authors state in lines 124-126 the presence of titanomagnetite in the residues of all experiments; thus, ilmenite and titanomagnetite exsolution cannot be excluded, given the inadequate evidence. I request that the authors provide definitive phase identification, otherwise, the pressure threshold for Nb depletion is not demonstrated.

Most of the rutiles are sub-micron, but after days of mapping out the samples by eye several relatively large rutiles have been found and they display their characteristic twins and with a beam of 15 kV we have obtained clean spectra of pure TiO_2 . We can thus confirm the presence of rutile. We now say this in **lines 146-150**.

The claim that garnet is unstable at 1.2 GPa and 1015 °C is unsupported; no 1.2 GPa experiment is reported in the SI. Perhaps the 1.8 GPa experiment should be re-run at 1.2 GPa, as it was at 1.4 GPa?

We ran a confirmation experiment (EPTgw20) at 1.2 and 1020 degrees and grew no garnet. The experiment generated melt in equilibrium with an amphibole, cpx and plag-rich residue. **We now highlight this confirmation experiment in line 140 of the main text and show images of the experiment with no garnet growth in the supplementary material (see lines 143-144 and 396-403 and new figure S2). The melt analyses and mass balance for this new sample is also added to the supplementary data table and the sample is plotted in Figure 2.**

Given that the 24-hour duration of the runs seems a little short, this might help clarify this issue. This is particularly notable because the phase equilibrium models of Johnson et al. (2017) predict garnet being stable down to a pressure of 0.8 GPa for the same general lithology (CF2, albeit an average of more samples).

24 hours is not short. There is always a trade off with experimental runs. The run has to be long enough to grow crystal phases of sufficient size that are in equilibrium; however, the longer the experimental run the more chance of hydrogen escape from the capsule that will alter the oxygen fugacity. It has long been recognised that 24 hours is a good time to grow large enough phases and to avoid damaging hydrogen loss. **We now make this point in lines 377-382.**

I find the choice of samples lacking negative Nb-Ta anomalies difficult to reconcile if the authors' goal was to perform experiments on a lithology representative of the broader CF2 suite.

We use **'type example'** 179879, which is representative of the Coonterunah group (CF2 suite). From Smithies et al. (2009) our sample (red dot) plots with the other Coonterunah samples (blue dots):

Smithies et al. (2009) use sample 179879 as a 'type example' of CF2 (see Smithies Table 1 below).
We now say our sample is representative in lines 102 and 105.

[REDACTED]

The manuscript acknowledges compositional diversity within the Coonterunah metabasalts, including pre-existing Nb troughs. As demonstrated in Johnson et al. (2017), average Coucal Basalt

has a negative Nb-Ta anomaly, so this calls into question whether the experiments are actually applicable to the stratigraphic unit.

Yes, the experiments **use a sample from the Coucal Basalts so they are applicable to the Coucal Basalts – it is the type example from Smithies et al. (2009)**. Some of the Coucal basalts have strong negative Nb anomalies and some don't. Our whole point is to stabilise rutile and generate a negative Nb-Ta anomaly. So, we use a sample that has high TiO₂ (rutile is pure TiO₂) so we can maximise the chance of saturating rutile and demonstrate the growth of a Nb-Ta anomaly. **We say this concisely in lines 111-115 and we are afraid that we don't know how to make this point any clearer to the reader.**

The Nb-Ta trough already existing in the average composition of the unit would suggest that, as a whole, rutile isn't necessary to generate a negative Nb anomaly.

We agree partly with the reviewer here. Rutile isn't necessary in samples with pre-existing Nb-Ta anomalies to generate granitoid melts with Nb-Ta anomalies. **We say this as our point (2) on line 273 and we are not sure how to make this clearer.** Basically, rutile is not stabilised at pressures <2.0 GPa. So, granitoid melts matching Pilbara plutonic rocks can be formed by either (1) direct partial melting of a subducting slab or foundering crust composed of Coonterunah-like basalts at ≥2.0 GPa to stabilise garnet and rutile in the residue and/or (2) generation of Coonterunah basaltic magmas with negative Nb anomalies that ascended and formed the Pilbara crust which subsequently re-melted ≤1.4 GPa to generate Pilbara granitoids.

The reduction of the Shaw (1970) relationship to $(C_i = C_0X)$ by folding F, P and D into a single constant X requires similar modal mineralogy and partitioning across different Coonterunah protoliths. That assumption is strong, and it is not validated.

The sample we use is representative of the group as a whole (see previous bivariate plots and Smithies Table). Therefore, the samples will have similar modal mineral assemblages and the partition coefficients will always be near-identical because the trace elements will adhere to Henry's law behaviour. **We make this point in a modified line 234.**

The modelling is then used to argue that a Nb-depleted source can reproduce a Nb trough at 1.4–1.6 GPa, which undercuts the headline claim that TTG-like Nb requires ≥2.0 GPa. I suggest the authors either experimentally test the model with at least one run on the Nb-depleted sample (e.g., 179865) at 1.4–1.6 GPa, or significantly soften the inference. As written, the logic is circular.

This is not circular, it is modelling using standard geochemical mass balance. We also do not 'undercut' a headline as we say specifically that Pilbara plutonic rocks can be formed by either (1) direct partial melting of a subducting slab or foundering crust composed of Coonterunah-like basalts at ≥2.0 GPa to stabilise garnet and rutile in the residue and/or (2) generation of Coonterunah basaltic magmas with negative Nb anomalies that ascended and formed the Pilbara crust which subsequently re-melted ≤1.4 GPa to generate Pilbara granitoids.

The authors argue that the granitic and granodioritic compositions of the experimentally-derived partial melts can be reconciled with natural observations because Paleoproterozoic tonalites and trondhjemites in the Pilbara "represent only a very small volume of the granitic complexes, whereas granites and granodiorites are the dominant silicic plutonic rock-type." This appears to be a misunderstanding. The granitic complexes of the EPT contain multiple supersuites, and the later ones, particularly from Emu Pool onwards, contain increasingly more granites and granodiorites. The granites in the younger Paleoproterozoic supersuites have been demonstrated to be the result of partial

derivation from re-melting of older TTG supersuites (e.g., Gardiner et al., 2017; Champion and Smithies, 2019; Hickman, 2021). The same goes for the granites in the Mesoarchean supersuites of the EPT. On the other hand, the Callina and Tambina supersuites, which are subordinate in area to the younger supersuites, are only 31% and 7% comprised of granite and granodiorite, respectively (using the compilation of Vandenburg et al., 2023, which incorporates all currently available analyses from the Pilbara). On the other hand, tonalites and trondhjemites comprise 30 and 31% of the Callina and Tambina supersuite samples (see attached Figure 1). While there might be some sampling bias here, owing to the sparsity of outcrop and the inability to sample significant portions of some of the domes due to heritage sites, the areal extents of granitoid lithologies in the two supersuites, as mapped by the Geological Survey of Western Australia, are consistent with granites being highly subordinate to tonalites and granodiorites (see attached Figure 2). While there are some examples of granites in these supersuites (e.g., the Homeward Bound Granite in the Callina Supersuite), these low-degree melts are relatively rare.

There are plenty of granite and granodiorites in the EPT. However, one needs to estimate proportions based on available geochemical data. The fact is that these more potassic compositions are a significant component, and they need to be explained. Some of the oldest rocks at 3.53 Ga are granodiorite. There is the North Pole monzogranite at 3.45 Ga., Plus all of the rhyolitic compositions in the eruptive sequences in the greenstone belts, from 3.53 (Coucal), 3.47 (Duffer) to 3.45-3.43 Ga (Panorama Fm). We are also afraid that the 'younger granodiorites forming by remelting the tonalites' is not supported by isotope data, at least to 3.2 Ga, and probably younger. Please see Kemp 2023 EPSL.

The authors imply that a wet environment would be best explained in a convergent margin. However, all that can really be shown here is that the fluid-present or fluxed melting of metabasalts indeed likely contributes to their production (e.g., Hernández-Urbe, 2024; Pourteau et al., 2020), but subduction is not a prerequisite for this style of intracrustal melting in the Archean (Hartnady et al, 2022), or in the Phanerozoic, for that matter (e.g., Weinberg and Hasalová, 2015). Numerous recent works have demonstrated that the dehydration of mafic-ultramafic lithologies in the crust can release sufficient water to produce TTGs (e.g., Hartnady et al., 2022; Hernández-Urbe, 2024; Pourteau et al., 2020; Tamblyn et al., 2023). Studies that infer a subduction-related petrogenetic origin for TTGs based on evidence for fluid-fluxed melting (e.g., Ge et al., 2023) only demonstrate that this melting process occurred within the crust. I'm not saying that TTGs can't be produced in subduction zones, more that they aren't diagnostic of the presence or absence of them.

We thank the reviewer for this comment and we agree with the statement. All we say in the manuscript is that, yes, the source has to be hydrous, but it also has to be, ultimately, deep to stabilise rutile and garnet. We do say that subduction and/or crustal drip and/or delamination tectonic environments (Lines 121, 280-282 and 289-290) are most applicable to either generate direct melts or the rocks for intraplate partial melting.

The crustal thickness estimates that the Authors invoke as evidence for subduction-like processes are based on La/Yb, which is only one geochemical parameter. How would this compare to another parameter, such as Eu/Eu* in zircon, or even better, a method that integrates multiple elemental and isotopic proxies (e.g., Luffi and Ducea, 2022)?

We use the modern combined method of Ce/Y and La/Yb to estimate a crustal thickness of 25 km (this is from a well renowned high-impact study that focuses specifically on Pilbara). The Luffi and Ducea method (which is a general technique) also shows a thickness of 25-30 km.

TTGs are, by definition, melts of pre-existing basaltic crust that have not interacted with mantle peridotite (e.g., Smithies et al., 2000; Moyen, 2020).

This is incorrect. TTG are a suite of Na-rich granitoid rocks classified by the IUGS using the Streckeisen diagram and also using the Barker An-Ab-Or plot. TTG have no petrogenetic mechanism as part of their definition.

Tonalites and trondhjemites formed directly by subduction-related processes would have to be derived from the slab, inevitably reacting with mantle wedge peridotite during ascent, thereby becoming high-Si adakites (and producing reciprocal residues that subsequently melt to produce sanukitoids) before they can be emplaced in the crust (Martin et al., 2009; Rapp et al., 2010; Smit et al., 2024; Smithies, 2000). I think the authors need to provide a mechanism with which TTGs generated from the slab would be able to ascend to the crust without contamination from the mantle if they want to pursue that interpretation.

Potential Pilbara TTG melts, derived from a slab, can easily ascend without incorporating mantle material (developing a 'mantle-signature') if (a) part of the slab is subducting at a flat angle, previous slab melts 'armour' ascending melt pathways into the lower crust to prevent/limit slab melt hybridisation and/or (c) ascending TTG magmas undergo fractional crystallisation to lower MgO, Ni and Cr contents. **We add this section to lines 300-304.**

The authors do not engage with recent work illustrating that many processes obfuscate the traditional interpretation/division of TTGs into LP-HP-MP categories, such as interstitial melt loss (Laurent et al., 2020), MgO content of the starting protolith (Johnson et al., 2017), mineral segregation (Kendrick and Yakymchuk, 2020; Kendrick et al., 2022), degree of melting and fractional crystallization (Smit et al., 2024), and melt hybridization (Hernández-Montenegro et al., 2021).

We are afraid this is not a review paper and it is beyond the scope of our study to discuss the obscure and unclear (as the reviewer states) interpretations of the rest of the community. We focus on studying the type example of the potential source of Pilbara TTG and then present the data.

The planetary angle and broad geodynamic statements are not anchored by new data in this study and are read as overreach. Plate tectonics is a very specific regime (see e.g., Cawood et al., 2022; Nebel et al., 2024). Neither drips/delamination nor subduction-like processes imply a global network of discrete plates separated by convergent, divergent, and transform boundaries with movements relative to one another governed by slab pull and minor amounts of ridge push. Such features can also occur in other types of non-plate tectonic regimes, such as sluggish lid (Lenardic, 2018), squishy lid (Lourenço et al., 2020) or lid-and-plate regime (Capitanio et al., 2019). I suggest keeping the focus on what the experiments demonstrate and moving speculative implications to a restrained paragraph.

We respectively disagree with this statement. If mafic crust is subducting or dripping (current drip models look very much like subduction zones) then new material has to be constructed somewhere. The surface of the Earth is not shrinking. So, if material is going down then it has to be growing somewhere else. This means there must have been constructive plate boundaries and these would also involve transform systems etc. Such a scenario sounds very similar to a proto-type of plate tectonics. **We modify lines 293-295 to say this.**

The authors state on line 120 the equilibration of the compositions at a NNO environment, but in the methods state "However, experiments cannot be buffered using an NNO assemblage". This seems

misleading to me and should be clarified in the main text, as non-specialists might think the experiments are being run at NNO.

Line now modified.

Line-by-line suggestions:

Line 72: I think a more detailed map figure might be warranted here; in its current state, it doesn't provide much context other than the location of the CF2 site.

Apologies, can we ask what details the reviewer wishes to see?

Line 79: Following up on my comment on line 66, I'd argue that a study like Vandenburg et al. (2023) deals more with the tectonic evolution of the craton than Francois et al. (2014) and would be a more appropriate reference here.

Reference changed.

Lines 89-90: Almost all the supersuites in the EPT are intruded by the Split Rock Supersuite, so this information isn't germane here.

Apologies, we don't know what the reviewer is suggesting here.

Line 93: Smithies et al. (2005- Whundo) isn't really an appropriate reference here. Perhaps you're thinking of "It started with a plume..." by Smithies et al. (2005)?

We thank the reviewer for this. Reference changed.

Line 112: Chowdhury et al. (2021) estimated a crustal thickness of ~50 km at 3.35 Ga for the Singhbhum Craton. See also Figure 6 in Chowdhury et al. (2025).

Thank you for the information. This supports our 1.4 GPa cut-off nicely.

Fig. 3: Specify the MORB normalization dataset; it is ambiguous as to which study the normalizing factors are from. Likewise, N-MORB is a surprising choice of normalizing factor for Archean rocks, as N-MORB did not exist during the Archean (Barnes et al., 2021). I think normalizing to primitive mantle would be a better choice. Fig. 3: The shaded areas denoted "East Pilbara TT" and "East Pilbara GG"; are they the full range of trace element data for compiled granitoids? They're so broad that they are not discriminating and don't really impart much meaningful information. Perhaps a better idea would be to plot the mean, median, or geometric means bounded by something like the 25th and 75th percentile values?

Figure 3 is converted to using a primitive mantle normalisation dataset from McDonough and Sun (1995). Also, if normalised values are discussed in the text, these have been converted into primitive mantle normalised values as well. Finally, mean values plotted.

Fig. 3: Trace element ratio-wise, melts from 179989 don't seem to be very representative of Pilbara granitoids.

We assume the reviewer means 179789. Our sample is representative (see previous comments).

Lines 189-190: The authors state “All of the 1.4 and 1.6 GPa partial melts have trace element patterns unlike those for East Pilbara tonalites and trondhjemites (TT) and East Pilbara granites and granodiorites (GG) (Fig. 3a).” Yet, the way that the figure is presented currently suggests that the partial melts actually do have trace element patterns similar to EPT granitoids.

The modified figure 3a now shows that the trace element patterns do match.

Line 203: I believe this is a misinterpretation once again of Hickman (2004), where the line about granitoids being “dominantly monzogranite and granodiorite” is in reference to the granitoids of the Pilbara Craton as a whole, rather than the EPT, let alone the Callina and Tambina Supersuites. As I have discussed above, the authors’ assertion that granites and granodiorites are “volumetrically dominant in the East Pilbara plutonic centres” is not germane to the central comparison, given that this volumetric dominance is the consequence of later magmatism incorporating an increasing amount of reworked components, rather than due to conditions during the generation of TTGs of the Callina and Tambina Supersuites.

We are studying the granitoid compositions derived from partial melting of the CF2 basalts, which could represent the source region for all the granitoids in the Pilbara Craton. As such, we discuss everything.

Lines 256-258: The majority of the evidence does not support this inference (Hickman et al., 2021, 2023), at least not until ca. 3.25 Ga, ca 200 Myr after emplacement of the Callina and Tambina Supersuites.

We are not taking sides here. This section is just to highlight the controversy surrounding the different models proposed to explain the formation of Pilbara. We have to present both sides of the argument.

Lines 273-276: One wouldn’t expect the eclogitic residue to be observed in outcrop. The TTGs were emplaced into the middle crust (e.g., Wiemer et al., 2018; Champion and Smithies, 2019), and there are no sections of lower crust exposed in the Pilbara. Given the extreme density contrast between purported eclogites and felsic melts, I don’t see how a scenario where TTGs carried eclogite fragments during migration from the lower to the middle crust would be physically possible (i.e., Stokes’ Law). I also cannot recall, to the best of my knowledge, an example of such a phenomenon occurring in Phanerozoic/modern localities associated with crustal delamination.

This is a good point by the reviewer. We modify this section. Please see lines 285-286.

Reviewer #3 (Remarks to the Author):

The results are promising and supportive of the conclusion, and can be accepted conditionally if my concerns listed below are thoroughly addressed.

In general, this study seems quite consistent (and similar) to another published study of the first author (Hastie et al., Nature Geoscience, 16, 816-821), which used a different starting material from a basaltic rock that of the southwestern Pacific Ontong Java Plateau; but for an unclear reason, this publication is not cited in this submission. I would suggest that the authors clearly emphasize the breakthrough of this submission compared to the previous one, if the starting compositions, the logic and design of experiments are quite similar.

This point is the same as point 1 from Reviewer 2. The previous study by Hastie et al. (2023) investigates a modern oceanic plateau source region (the Ontong Java Plateau: 1187-8 and 1187-10) as analogues of the early Earth's mafic crustal surface. The compositions of the old material (1187-8 and 1187-10) and the starting mixes here (EPT) are shown below:

Sample	SiO ₂	TiO ₂	Al ₂ O ₃	FeO(t)	MnO	MgO	CaO	Na ₂ O	K ₂ O	P ₂ O ₅
EPT2019	48.069	2.17	14.855	13.75	0.24	7.72	9.68	2.42	0.96	0.14
EPT2021	47.822	2.14	14.639	14.44	0.24	7.66	9.59	2.38	0.94	0.14
EPT2024	48.10	2.07	15.58	13.84	0.25	7.59	9.17	2.32	0.95	0.13
1187-8	49.22	0.75	14.87	9.76	0.17	9.89	12.39	1.64	0.09	0.06
1187-10	49.89	0.73	14.53	9.85	0.16	10.20	12.77	1.73	0.10	0.06

Of the ten major elements, 8 are completely different in the new starting mix (highlighted green) and the trace element concentrations are an order of magnitude different. **We take full responsibility for this confusion and now modify the text to make it clear that the previous studies are not related (Lines 97-101).**

Another concern is comparing the experimental results with the conflicting data from thermodynamic modeling (Johnson et al., 2017, Nature 543, 239-242), which used almost the same starting composition and target TTG for testing their results. A reasonable discussion may solve or release the puzzle of readers.

We are afraid that we are not sure we can do this now. There are problems with the older thermodynamic models because equations of state behind these thermodynamic packages had algebraic errors and they cannot predict accessory phase stability, please see:

Green, Holland, Powell, Weller, Reil. 2025. Corrigendum to: Melting of Peridotites through to Granites: a Simple Thermodynamic Model in the System KNCFMASHTOCr, and, a Thermodynamic Model for the Subsolidus Evolution and Melting of Peridotite. Journal of Petrology. <https://doi.org/10.1093/petrology/egae079>.

Specifically, many of these previous modelling studies rely on the stability of the accessory phase 'rutile' in relatively shallow crustal environments (<45 km depth) in order to form magmas that solidified into the oldest continental crust. However, the thermodynamic models are now heavily suspect with regards to accessory phase stability.

However, our study provides direct experimental evidence for melting the Pilbara source region. Such experiments underpin thermodynamic models. Thus, if the experiments contradict the models, this is because the models, for some reason or another, are not generating accurate results.

My third concern is that, accepting that a pressure of >1.4 GPa (>45 km) is needed for the formation of TTG rocks, how to conclude solely that a subduction-like tectonic environment has been the only possibility, i.e., how to exclude crustal drips/delamination model which is similarly consistent with such a high pressure (as the authors put in the abstract).

We apologise we meant to emphasise that a drip/delamination model is also viable. We modify what we say now and mention that subduction and/or crustal drip and/or delamination tectonic environments (**Lines 121, 280-282 and 289-290**) are most applicable to either generate direct melts or the rocks for intraplate partial melting.

Response to referee 2 for manuscript NCOMMS-25-59996-T

From the perspective of a geochemist, I still believe it substantially overlaps with Hastie et al. (2023), Law et al. (2024) and Law and Hastie (2025), without a clear additional and substantial advance, beyond anything that will be evident to readers other than those from a niche portion of the experimental petrology community.

This statement by referee 2 is a factual error. Firstly, Law et al. (2024) is a whole rock geochemical study of Icelandic rocks and involves no experimentation (<https://www.nature.com/articles/s43247-024-01513-5>). Further, we demonstrated in our first response that the starting composition and results of this manuscript are completely different to the older studies. We clearly showed that our manuscript uses a composition that is hugely different to the older studies with regards to both major and trace elements – **it would be difficult to find two compositions further apart!** Referee 2 chose to ignore this completely. However, to repeat ourselves:

This study is, to our knowledge, the **first** and **only** major and trace element experimental study of an ancient metabasic source region from Pilbara that has starting compositions similar to 'island arc basalts (Samples 179789, EPT2019, EPT2021, EPT2024). The two starting compositions (the ones from Hastie and Law – 1187-8 and 1187-10) are **completely and utterly different with regards to both major and trace element concentrations:**

Sample	SiO ₂	TiO ₂	Al ₂ O ₃	FeO(t)	MnO	MgO	CaO	Na ₂ O	K ₂ O	P ₂ O ₅
EPT2019	48.069	2.17	14.855	13.75	0.24	7.72	9.68	2.42	0.96	0.14
EPT2021	47.822	2.14	14.639	14.44	0.24	7.66	9.59	2.38	0.94	0.14
EPT2024	48.10	2.07	15.58	13.84	0.25	7.59	9.17	2.32	0.95	0.13
1187-8	49.22	0.75	14.87	9.76	0.17	9.89	12.39	1.64	0.09	0.06
1187-10	49.89	0.73	14.53	9.85	0.16	10.20	12.77	1.73	0.10	0.06

Of the ten major elements, 8 are completely different in the new starting mix (highlighted green) and the trace element concentrations are an **order of magnitude** different. Finally, Pilbara is at the **height** of controversy regarding the evolution of the early continental crust so the results of this study will be of great interest to the scientific community and will be highly sought after.

This isn't the first experimental study with major and trace elements measured in tandem (see e.g., Xiong et al., 2005). For experimental studies using Archean rocks (which inherently do not have modern oceanic plateau-like compositions), this is neither the first one to measure major elements (Rapp et al., 1991 and

Rapp and Watson, 1995 use WR-40, a Neoarchean amphibolite from the Wyoming Craton), nor is it the first to measure major and trace elements in tandem (Adam et al., 2012 used Eoarchean Nuvvuagittuq amphibolites from the Superior Craton as their starting materials). These may be compositionally different from the Coonterunah metabasalts, but it does void the statement of priority.

This whole statement is factually misleading. Specifically:

1. Xiong et al. 2005 used a MORB starting mix and nothing from an ancient source; thus, further demonstrating that our starting mix is unique. Also, they only measured Ta, Nb, Hf, Zr, La, Sm, Lu, Rb, Cs, and Sc, but unlike our analyses that are natural and specifically investigating TTG genesis, Xiong doped their experiments with 100 ppm of each element. This is because they were not replicating TTG genesis *sensu stricto*, they were exploring elemental partitioning.
2. **For the Rapp study, referee 2 proves our point.** Rapp used a sample that was 2.7-2.6 billion years old and has an E-MORB composition. In contrast, our starting composition is nearly a billion years older and has an island arc-like basaltic composition – completely different.
3. Adam did experiment on a sample from Superior, but it was a geochemically unique boninite composition that is very distinct to many of the rocks found in ancient Greenstone belts – completely different to our composition as well.

Basically, our study is the first natural (undoped) major and trace element experimental study using an ancient island arc basalt-like metabasic rock that are ubiquitous in Hadean-Palaeoarchean terranes. To make this clear we have highlighted the novelty of our approach and results by re-writing lines 100-113. We have also added all previous experiments in Table S1 for comparison.

One glaring issue that I don't believe I caught in the original manuscript is that the experiments in this study do not produce magmas with trondhjemitic compositions, which are typically associated with the high-pressure subgroup of TTG series magmas inferred to be produced by a plagioclase-free rutile and garnet-bearing source (e.g., Hoffmann et al., 2019; Laurent et al., 2024; Moyen and Martin, 2012; Moyen, 2020). This begs the question whether the results of the experiments can genuinely be considered representative of high-pressure TTGs, let alone those in the East Pilbara and thus calls into question the authors' inferences based on these results.

Most direct partial melts of metabasic sources are tonalites and granodiorites – so our results are perfectly sound. **In fact, it would be more irregular to get trondhjemites!** Trondhjemites are sub-ordinate to the two former rock types. So, it is no surprise that the experimental liquids are not trondhjemitic. Trondhjemites are dominantly formed by the fractional crystallisation of tonalite and granodiorite magmas (see Barker 1979 for an original take on this and a good summary in Drummond et al. 1996).

Another related issue involves the discrepancies between the experimental results and data from thermodynamic modelling (e.g., Johnson et al., 2017). I'm not satisfied with the authors' response to Reviewer 3's comment on this issue. The issues arising from the Magmatic NCKFMASHTOCr dataset of Holland et al. (2018) are unlikely to have a substantial impact on the stability of accessory phases. While the authors cite the corrigendum of this study as evidence for something being faulty with thermodynamic modelling, this dataset is entirely irrelevant to the results of Johnson et al. (2017) or White et al. (2017), as these studies used the metabasite NCKFMASHTO dataset of Green et al. (2016). Furthermore, the issues that the authors misattribute as being the main contributors to the discrepancy with their results have been fixed, so it is an easy (30 minutes tops) and worthwhile endeavour for the authors to compare their results with the most up-to-date thermodynamic modelling in MAgEMin or Thermocalc. It is unlikely that the thermodynamic datasets, which are constantly updated and informed by the results of numerous experiments, are incorrect, and I am not convinced of this on the basis of one experimental study.

Apologies, but again, this whole statement is factually incorrect. Specifically:

1. The issues with the thermodynamic software have a huge impact on the stability of accessory phases. The authors of the correction paper state that accessory phase stability has a large uncertainty in the modelling outputs and the models for accessory phases are particularly imprecise. We have taken the liberty of showing this directly in a screen shot of the paper below (see highlighted). The correction shows that the problems with the accessory phases cannot be 'fixed' because their Gibbs free energy contributions are too small at present. Also, coincidentally, in the same highlighted section, the correction highlights modelling discrepancies between older tonalite models and more recent results. So, older tonalite genesis models are to be viewed with extreme suspicion.

[REDACTED]

2. Additionally, Referee 2 wishes to believe thermodynamic modelling results over experimental evidence – this is odd in itself as experiments are needed to refine these models. So, asking for a model to prove an experiment is highly irregular and is not supported by basic scientific principles.

The authors have removed their original argument towards the end of the main text about eclogite entrainment, but now jumps from inferences on TTGs made based on experimental results to unsubstantiated claims about the origin of the mafic protoliths employed in this study. This reads equally unsatisfactorily, given that this is unsubstantiated, beyond the scope of this study, and disregards substantial evidence to the contrary.

It now obvious to us that trying to present arguments for both subduction and intracrustal processes is causing some concern. Thus, on advice from the Nat Comm editorial team we have slimmed this section down and removed sentences in lines 178-182, 270-281, 298-305 and 320-322. We advocate a complex

tectonic environment on the early Earth with both subduction and intracrustal processes (other studies pick one or the other) – this also makes our study unique. We now highlight this better in the text and we have even modified the title to include both models.

Granitoids:

TTGs can be IUGS-defined tonalites, trondhjemites and granodiorites. However, the “TTG series”, which I assume the authors are employing as the basis of their arguments, has a specific connotation in the Archean community (e.g., Moyen and Martin, 2012; Moyen, 2020; Spencer, 2025). Attributing all compositionally trondhjemitic, tonalitic and granodioritic compositions to the TTG series reflects a lack of engagement with the literature and the commonly used criteria that separate the TTG series from other Archean granitoids. Asserting that tonalites, trondhjemites and granodiorites are compositionally identical to the TTG series as defined by seminal studies/articles such as Smithies and Champion (2000), Martin et al. (2005), Moyen and Martin (2012), Moyen (2020) and Laurent et al. (2024) without actively employing such criteria, perpetuates a misconception in the broader igneous community that the two are analogous. The authors should perhaps consider the alternative models of Laurent et al. (2021) and Smit et al. (2024) as the processes outlined in this study have a direct impact on Nb contents, which is traditionally a proxy for residual rutile in TTGs.

Apologies, but again, this whole statement is factually incorrect. TTG are strictly defined under IUGS guidelines. All serious studies related to TTG use these guidelines. In our manuscript we follow IUGS definitions.

Geological

observations:

As I discussed in my comments on the previous round of reviews, granites and granodiorites are not the dominant silicic plutonic rock types for the Callina and Tambina Supersuites. While there are occurrences of granites in these supersuites as pointed out by the authors, and although I agree that some of these > 3.5-3.4 Ga granites are likely produced by low-degree melting of amphibolites, in reality, these are volumetrically subordinate. To use the example that the authors provide as evidence, the North Pole Monzogranite is a relatively small intrusion. Likewise, rhyolites are not particularly germane to the argument as many of these are not genetically related to TTGs and do not share the same source (e.g., Smithies et al., 2019), with some exceptions, including the Coucal “TTG flow” of Smithies et al. (2009).

In their rebuttal, the authors state, “one needs to estimate proportions based on available geochemical data.” I did just that in the original round of reviews, providing two figures using two methods to demonstrate that granites and granodiorites are subordinate to tonalites and trondhjemites in the Callina and Tambina Supersuites. I have reattached the two figures from the original round of reviews and recommend that the authors reread my original comments, as their response has been unsatisfactory.

I’m familiar with Kemp et al. (2023), and it is certainly an important study. However, it doesn’t exist in a vacuum, especially with respect to Hf isotope data in granitoids and has its shortfalls. For one, Kemp et al. (2023) do not consider the work of Gardiner et al. (2017, 2021). Additionally, samples from all terranes are presented together in the ϵHf -Age diagrams rather than the samples from each terrane separately.

Because the different terranes of the Pilbara evolved as independent entities, this can lead to some spurious inferences when grouped on the same diagrams. I believe this could explain the authors’ incorrect assertion that “younger granodiorites forming by remelting the tonalites’ is not supported by isotope data, at least to 3.2 Ga” in response to my comment about the presence of reworked components in granitoids from some of the 3.32-3.29 Ga Emu Pool Supersuite intrusions and many of the 3.27-3.22 Ga Cleland Supersuite intrusions. To demonstrate the presence of this component from 3.3 Ga onwards, I’ve plotted all available Hf isotope in zircon data for magmatic grains (within $\pm < 5\%$ discordance) from East Pilbara Terrane granitoids versus time in Fig. 3. As can be seen, both sample mean ϵHf and individual zircon grain ϵHf begins to decrease around 3.3 Ga (coinciding with the Emu Pool Supersuite) and gets especially pronounced and subchondritic

around 3.27 Ga (coinciding with the Cleland Supersuite). This shift is best reconciled with the presence of a reworked component. This isn't to say that all granitoids from these supersuites contain the component, as there are clearly intrusions with suprachondritic ϵ_{Hf} , but many do.

Other lines of evidence exist in addition to the changes in granitoid geochemistry towards more potassic compositions (Vandenburg et al., 2023) and the outlined Hf isotope systematics. Gardiner et al. (2021) also provide further evidence for reworked components based on the presence of xenocrysts in the central portions of Emu Pool Supersuite intrusions, suggesting inheritance occurred at depth in their sources, rather than due to wall rock entrainment. Furthermore, oxygen isotope data presented by Smithies et al. (2021) and Johnson et al. (2022) from granitoids of the Emu Pool and Cleland Supersuites are best reconciled with the presence of a reworked component. Altogether, evidence from the literature most parsimoniously suggests that the latter two Paleoproterozoic granitoid supersuites of the East Pilbara increasingly incorporated a reworked component, as opposed to being produced by low-degree melting of an amphibolite source. Therefore, the results of this study are not entirely relevant to the <3.3 Ga granitoids of the Pilbara and the authors should be more explicit to state that these results apply mostly to the > 3.3 Ga supersuites.

I understand that this is not a review paper, but the authors need to take a more holistic approach when making inferences based on their results and account for the existing literature from the Pilbara. The four Paleoproterozoic granitoid supersuites of the East Pilbara Terrane were emplaced contemporaneously with the ultramafic-felsic lavas of the Pilbara Supergroup. If subduction-like processes produced the TTGs, then one would expect that the Pilbara Supergroup lavas would bear geochemical signatures consistent with such a tectonomagmatic setting, given their close spatio-temporal association. However, there is no evidence for such an origin for the lavas of the Pilbara Supergroup, and they are better explained by variably crustally contaminated volcanism in a mafic plateau setting above a mantle upwelling (e.g., Brown et al., 2024; Hasenstab et al., 2021; Maier et al., 2009; Murphy et al., 2021; Nebel et al., 2014; Puchtel et al., 2022; Smithies et al., 2005; Tynpel et al., 2021). Thus, as concluded by numerous studies using field, geochronological, structural, and geochemical data, geological context of the EPT in the Paleoproterozoic remains most parsimoniously explained by processes not diagnostic of plate tectonics (e.g., Brown et al., 2024; Campbell and Davies, 2017; Gardiner et al., 2017; Hasenstab et al., 2021; Hawkesworth and Kemp, 2021; Hickman, 2021, 2023; Johnson et al., 2017, 2022; Roberts et al., 2022; Smithies et al., 2005a, 2009, 2021; Van Kranendonk et al., 2015, 2019; Wiemer et al., 2018). While experiments can offer interpretations and possible explanations of data, experimental results do not constitute proof of the operation of a given process.

We thank referee 2 for the time they have taken to write up this background. **Unfortunately, most of this extensive section is not relevant. Specifically:**

3. We **do not** say that granites and granodiorites are not the dominant silicic plutonic rock types for the Callina and Tambina Supersuites.
4. Referee 2 says we say that 'younger granodiorites forming by remelting the tonalites'. I am afraid we **do not say this** in the manuscript.
5. The information deals with much younger rocks <3.25 Ga. Our study relates to rocks and processes 250 million years before this at 3.52 Ga.
6. Also, we repeat that: it now obvious to us that trying to present arguments for and against both subduction and intracrustal processes is causing some concern. Referee 2 is very supportive of the intracrustal model, which is fine. But, referee 2 must realise that the studies involving subduction as a mechanism give equally robust defences of their evidence. Thus, on advice from the Nat Comm editorial team we have slimmed our tectonic sections down and removed sentences in lines 178-182, 270-281, 298-305 and 320-322. We have even modified the title to include both models.

Geodynamics:

The planetary angle and broad geodynamic statements are not anchored by new data in this study and still read as overreach; the authors' response to my original comment on this matter does not rectify this issue.

Again, we repeat that: it now obvious to us that trying to present arguments for and against both subduction and intracrustal processes is causing some concern. Referee 2 is very supportive of the intracrustal model, which is fine. But, referee 2 must realise that the studies involving subduction as a mechanism give equally robust defences of their evidence. Thus, on advice from the Nat Comm editorial team we have slimmed our tectonic sections down and removed sentences in lines 178-182, 270-281, 298-305 and 320-322. We advocate a complex tectonic environment on the early Earth with both subduction and intracrustal processes (other studies pick one or the other) – this also makes our study unique. We now highlight this better in the text and we have even modified the title to include both models.

As I said in the original round of reviews, plate tectonics is a very specific type of geodynamic regime. To quote Stern and Gerya (2018), plate tectonics is defined as “a theory of global tectonics powered by subduction in which the lithosphere is divided into a mosaic of strong lithospheric plates, which move on and sink into weaker ductile asthenosphere. Three types of localized plate boundaries form the interconnected global network: new oceanic plate material is created by seafloor spreading at mid-ocean ridges, old oceanic lithosphere sinks at subduction zones, and two plates slide past each other along transform faults. The negative buoyancy of old dense oceanic lithosphere, which sinks in subduction zones, mostly powers plate movements.”. The return of mafic crust to the mantle (either through dripping or subduction) would, of course, be accompanied by rifting elsewhere to replace the volume of crust returned to the mantle, but it need not comprise a global, interlinked network of ridges, transform faults and subduction zones. As shown by multiple numerical modelling studies (e.g., Capitanio et al., 2019, 2020, 2022; Lenardic, 2018; Lourenço et al., 2020), this does not necessarily imply plate tectonics. If it did, then Venus would also be considered to have a “proto-type of plate tectonics” (e.g., Byrne et al., 2021; Capitanio et al., 2024; Gillmann et al., 2025), thereby voiding part of this pitch aimed at a more general scientific audience.

To repeat ourselves again, it now obvious to us that trying to present arguments for and against both subduction and intracrustal processes is causing some concern. Referee 2 is very supportive of the intracrustal model, which is fine. But, referee 2 must realise that the studies involving subduction as a mechanism give equally robust defences of their evidence. Thus, on advice from the Nat Comm editorial team we have slimmed our tectonic sections down and removed sentences in lines 178-182, 270-281, 298-305 and 320-322. However, the models cited are done in two dimensions. It is geodynamically implausible to have spreading and subduction localised to small areas of the planet – it just doesn't make sense at all. We advocate a complex tectonic environment on the early Earth with both subduction and intracrustal processes (other studies pick one or the other) – this also makes our study unique. We now highlight this better in the text and we have even modified the title to include both models.

Line-by-line

comments:

Line 34: What's meant by "continental" here? Does this refer to the UCC, MCC, LCC or the bulk CC? These all mean different things and have different implications.

Line re-worded.

Line 42: Absence? They're certainly rare, but Hadean xenocrystic and detrital zircons have been found in multiple cratons (see e.g., Condie, 2019 for a brief review).

Line re-worded.

Line 100: For metabasites without oceanic plateau-like compositions, this isn't the first experimental study with major and trace elements measured in tandem (see e.g., Xiong et al., 2005). For experimental studies using Archean rocks (which inherently do not have modern oceanic plateau-like compositions), this is neither the first one to measure major elements (Rapp et al., 1991 and Rapp and Watson, 1995 use WR-40, an Archean amphibolite from the Wyoming Craton), nor is it the first to measure major and trace elements in tandem (Adam et al., 2012 used Nuvvuagittuq amphibolites as their starting materials). These may be compositionally different from the Coonterunah metabasalts, but it does void the statement of priority.

Statement addressed above.

Line 110-111: Sample 179789 does, in fact, show evidence for crustal contamination by older silicic crust, as demonstrated in the SI of Johnson et al. (2017), so stating that the Coonterunah metabasalts "show no evidence that their parental magmas have been contaminated with older silicic crust" is inaccurate.

Smithies et al 2009 and the isotope data in Kemp et al 2023 suggests otherwise.

Lines 167-168: See discussion above about granitoids and geological observations

Statement addressed above.

Fig. 3: Given the above comments, I recommend the authors only plot > 3.32 Ga granitoids from the East Pilbara as shaded areas on their primitive mantle normalized extended trace element diagrams. Including the younger granitoids will invariably lead to spurious comparisons.

This is the case anyhow.

Lines 214-215: Again, see the discussion above about granitoids and geological observations. Each of the granitoid complexes in the East Pilbara is comprised of sets of composite intrusions spanning > 700 Myr of Earth's history and if one looks at a geological map of the area, it is evident that the < 3.32 Ga intrusions are more widespread. I believe this is a misinterpretation once again of Hickman (2004), where the line about granitoids being "dominantly monzogranite and granodiorite" is in reference to the granitoids of the Pilbara Craton as a whole, rather than the EPT, let alone the Callina and Tambina Supersuites. As I have discussed above, the authors' assertion that granites and granodiorites are "volumetrically dominant in the East Pilbara plutonic centres" is not germane to the central comparison, given that this volumetric dominance is the consequence of later magmatism incorporating an increasing amount of reworked components, rather than due to conditions during the generation of TTGs of the Callina and Tambina Supersuites.

Statement addressed above.

Lines 284-288: Not all TTGs are derived from eclogites, some are derived from amphibolites, as demonstrated by field evidence (e.g., Kendrick et al., 2024; Pourteau et al., 2020). Ascribing the entire source to eclogites is

inaccurate, especially given that high-pressure group TTGs are not as common as low-pressure group TTGs in the Pilbara (e.g., Champion and Smithies, 2019; Vandenburg et al., 2023).

Section removed to tone down tectonic discussion.

Line 289: This study does not show that a wet and deep tectonic environment is required to produce the Nb anomalies in the F2 Coonterunah basalts. This is granitoid petrology paper, not one on the mantle source of CF2. For what it's worth crustal contamination better explains many of the Nb anomalies in the CF2 basalts (e.g., Brown et al., 2024; Johnson et al., 2017). Crustal contamination can be quite cryptic in many cases as well (see e.g., Vite-Sánchez et al., 2024).

Section removed to tone down tectonic discussion.

Lines 296-298: In my opinion, this is overly speculative, lacks proper substantiation and diverts from the core message of this study.

Section removed to tone down tectonic discussion.

Lines 302-303: If subduction is occurring at a flat angle, then doesn't that contradict the statement in line 298 about the existence of steep subduction with a mantle wedge?

Line re-worded.

Lines 304-305: Fractional crystallization does not remove the mantle signature in modern adakites, even on the rhyolitic side of things, so I don't see how that would be the case in the Archean with TTGs. If there is subduction, it is unlikely for there to be a mantle wedge, not only because of the aforementioned considerations (see also Smithies, 2000; Martin et al., 2005), but also because all that armouring required to prevent mantle signatures in the TTGs would ultimately have to melt, producing sanukitoids and metasomatized mantle-derived mafic lavas, which are conspicuously absent from the East Pilbara Terrane.

This statement is false.

Lines 306-311: Wrt the ability of subduction-like processes to emit gases into the atmosphere, the emplacement style of the volcanoes is a critical question. The geological evidence is overwhelmingly in favour of subaqueous emplacement (e.g., Flament et al., 2008; Hickman, 2023) in most places, such that the hydrostatic pressure of the overlying oceanic water column affected the nature of emitted gases (e.g., Gaillard et al., 2011). This is consistent with multiple lines of evidence for very low sulfate concentrations in the Archaean ocean. There are many more related pieces of evidence regarding Archaean atmosphere-hydrosphere-lithosphere interaction that should be considered if trying to pursue this route (e.g., Kamber and Ossa-Ossa, 2025).

Response to referee 4 for manuscript NCOMMS-25-59996-T

We thank Reviewer 4 for their thorough and insightful review that has improved our manuscript. We now respond to the specific suggestions by review 4 below:

(1) A clearer definition of the specific time-frame the study is relevant to.

The title states 'Early Archaean', the abstract mentions 4.3-3.5 Ga and 4.0-3.5 Ga time periods, lines 83–93 appear to indicate the study is specifically focusing generation of the Callina and Tambina supersuites (~3.5-3.4 Ga), and several places in the text mention Eoarchean-Paleoarchean or Hadean-Paleoarchean crustal formation. Do the experimental results only pertain to the generation of Paleoarchean TTGs in the East Pilbara Terrane? If so, this could be stated more explicitly. Note that the Eoarchean record of the East Pilbara Terrane is sparse so line 77 needs amending (the Paleoarchean might be extensive, but the Eoarchean is only known from rare enclaves and detrital zircon and apatite grains). It is possible that the composition and crust-forming processes of the Eoarchean crust of the East Pilbara Terrane differ from the preserved Paleoarchean crust (compare Hickman, 2023, Archean Evolution of the Pilbara Craton and Fortescue Basin and Kharkongor et al. 2025, Geology). This point is also true for other Archean terranes where any Hadean-Eoarchean record is primarily preserved as detrital or xenocrystic zircon grains. There is no consensus on the composition of the original host rocks of Hadean-Eoarchean zircons (a quick survey of recent literature on Hadean zircon trace element chemistry includes estimates ranging from peraluminous granites, andesites, mafic rocks, and TTGs). As written the paper implies that the Paleoarchean TTGs of the EPT are representative of all early continental crust (including Eoarchean-Hadean). I think this is a reasonable assertion but the debate regarding the composition and origin of Eoarchean-Hadean crust and the possibility that it differed from the more widely preserved Paleo-Mesoarchean crust should be acknowledged.

We apologise for the confusion with the timings. We have now tidied up the text. 4.0 is changed to 4.3 in the abstract and 'Eoarchaeal' has been removed from lines 79, 122, 201 and 275 to show that we only reference Palaeoarchaeal TTG and the Palaeoarchaeal in general.

(2) The discussion of lithospheric recycling on the early Earth could be framed in the context of recent geodynamic modelling studies.

Inferring Early Earth geodynamic settings from geochemical data is highly contested and non-unique. I recognise that the authors have already made a significant effort to tone-down this aspect of the manuscript during earlier reviews. However, the inferences of the geodynamic setting of early Archean crust formation could be strengthened by discussing this aspect of the study in the context of recent geodynamic modelling. The study advocates for deep subduction and 'proto' or 'primitive' plate tectonics to generate Paleoarchean TTGs in the East Pilbara Terrane. 'Proto plate tectonics' and 'primitive plate tectonics' are nebulous terms and there have been recent efforts to clearly define the difference between modern plate tectonics and the geodynamic environments operating on the early Earth (see Cawood et al. 2022 for an accessible summary). Numerical and thermomechanical modelling also highlight important differences in geodynamic settings on a hotter early Earth compared to modern plate tectonics. These studies recognise several possible geodynamic modes on early Earth, for example, stagnant lid, squishy lid, mobile lid modes.

We have further 'toned down' the argument and referenced a few geodynamic models. We add sentence 274 to highlight support for co-existing tectonic models that we advocate with subduction and intracrustal processes. Also, we recognise that we shouldn't use 'proto' and 'primitive' so we have removed the term 'proto'.

Subduction or lithospheric dripping are cited as possible mechanisms for generating high-pressure TTG melts but the authors lean in favour of subduction and a plate tectonic like regime (e.g., in the title and abstract). Both these processes are consistent with 'squishy' or 'mobile' lid tectonic modes. Many readers may find these definitions more palatable than referring explicitly to plate tectonics. In my opinion, using

experimental data to demonstrate a diversity of melt-generating environments on the early Earth is an important and impactful finding alone. Describing the geodynamic setting of melting might be an issue of semantics, but at least using the more recent definitions outlined above provides a common framework for communicating the geodynamic setting the authors are envisaging.

We have indicated plutonic squishy lid terminology in lines 32, 63, 273. For mobile lids, I think we still prefer the terms 'subduction' and 'primitive plate tectonics'.

Minor points:

Inconsistent naming of the study area. The study area is referred to as 'Pilbara terrane' , 'northern Pilbara craton' , 'East Pilbara'. The study focuses on the EAST PILBARA TERRANE, which is a terrane within the Pilbara Craton. This should be updated throughout the text.

The terminology has been updated throughout the text. Apologies for any confusion.

Areal extents of granitoids as mapped for the Callina and Tambina Supersuites

Feldspar triangle (O'Connor 1965)

Figure 1. CIPW-normative feldspar classification ternary of O'Connor (1965) of Callina and Tambina Supersuite granitoids with overlain density contours, constructed using the whole rock geochemical database of Vandenburg *et al.* (2023), which provides all currently available geochemical analyses of granitoids from the Pilbara. The Callina and Tambina supersuites, which are subordinate in area to the <3.4 Ga supersuites, are only 31% and 7% comprised of granite and granodiorite, respectively. On the other hand, tonalites and trondhjemites comprise 30% and 31% of the Callina and Tambina supersuite samples, respectively.

Areal extents of granitoids as mapped for the Callina and Tambina Supersuites

Figure 2. Proportions by area of granitoid lithologies that comprise the Callina and Tambina supersuites, calculated using the 1:100 000 State interpreted bedrock geology of Western Australia shapefiles. Individual units were reclassified based on lithology, and their areas were calculated in ArcGIS Pro. This method provides an alternative way of calculating the proportion of lithologies as a workaround for the

influence of sampling bias that arises from the sparsity of outcrop and sampling restrictions in Aboriginal heritage sites. The results shown here are consistent with granites being highly subordinate to tonalites and granodiorites.

Figure 3. Plot of ϵ_{Hf} versus time for individual magmatic zircon grains (using $^{207}\text{Pb}/^{206}\text{Pb}$ dates) and sample averages (using sample ages) from Paleoproterozoic granitoids of the East Pilbara Terrane. 2σ errors are plotted and only grains with $\pm < 5\%$ discordance are considered, both individually and in calculating average ϵ_{Hf} . The data includes all currently available data from the literature: Buzenchi *et al.* (2022), Gardiner *et al.* (2017, 2021), Kemp *et al.* (2023), Lu *et al.* (2022), Petersson *et al.* (2019a,b, 2020), and Salerno *et al.* (2021).

References

Buzenchi A., Moreira H., Bruguier O. and Dhuime B. (2022) Evidence for Protracted Intracrustal Reworking of Palaeoproterozoic Crust in the Pilbara Craton (Mount Edgar Dome, Western Australia). *Lithosphere* **2022**.

Gardiner N. J., Hickman A. H., Kirkland C. L., Lu Y., Johnson T. and Zhao J. X. (2017) Processes of crust formation in the early Earth imaged through Hf isotopes from the East Pilbara Terrane. *Precambrian Res.* **297**, 56–76.

Gardiner N. J., Mulder J. A., Nebel O., Kirkland C. L. and Johnson T. E. (2021) Palaeoproterozoic TTGs of the Pilbara and Kaapvaal cratons compared; an early Vaalbara supercraton evaluated. *South African J. Geol.* **124**, 1–16.

Geological Survey of Western Australia (2022) 1:100 000 State interpreted bedrock geology of Western Australia, April 2022 update: Geological Survey of Western Australia, digital data layer, www.dmirs.wa.gov.au/geoview.

Compilers of geology: OA Blay, HN Cutten, HM Howard, TJ Ivanic, D Martin, R Quentin de Gromard, A Riganti, C Spaggiari, I Zibra

Kemp A. I. S., Vervoort J. D., Petersson A., Smithies R. H. and Lu Y. (2023) A linked evolution for granite-greenstone terranes of the Pilbara Craton from Nd and Hf isotopes, with implications for Archean continental growth. *Earth Planet. Sci. Lett.* **601**, 117895.

Lu Y., Wingate M. T. D., Smithies R. H., Gessner K., Johnson S. P., Kemp A. I. S., Kelsey D. E., Haines P. W., Martin D. M., Martin L. and Lindsay M. (2022) Preserved intercratonic lithosphere reveals Proterozoic assembly of Australia. *Geology* **50**, 1202–1207.

O'Connor, J.T. (1965) A classification for quartz-rich igneous rocks based on feldspar ratios. *U.S. Geological Survey Professional Paper* **525**, 79–84.

Petersson A., Kemp A. I. S., Hickman A. H., Whitehouse M. J., Martin L. and Gray C. M. (2019) A new 3.59 Ga magmatic suite and a chondritic source to the east Pilbara Craton. *Chem. Geol.* **511**, 51–70.

Petersson A., Kemp A. I. S. and Whitehouse M. J. (2019) A Yilgarn seed to the Pilbara Craton (Australia)? Evidence from inherited zircons. *Geology* **47**, 1098–1102.

Petersson A., Kemp A. I. S., Gray C. M. and Whitehouse M. J. (2020) Formation of early Archean Granite-Greenstone Terranes from a globally chondritic mantle: Insights from igneous rocks of the Pilbara Craton, Western Australia. *Chem. Geol.* **551**, 119757.

Salerno R., Vervoort J., Fisher C., Kemp A. and Roberts N. (2021) The coupled Hf-Nd isotope record of the early Earth in the Pilbara Craton. *Earth Planet. Sci. Lett.* **572**, 117139.

Vandenburg E. D., Nebel O., Smithies R. H., Capitanio F. A., Miller L., Cawood P. A., Millet M.-A., Bruand E., Moyen J.-F., Wang X. and Nebel-Jacobsen Y. (2023) Spatial and temporal control of Archean tectonomagmatic regimes. *Earth-Science Rev.* **241**, 104417.